# FTA: Stealthy and Adaptive Backdoor Attack with Flexible Triggers on Federated Learning

## Abstract

Current backdoor attacks against federated learning (FL) strongly rely on universal triggers or semantic patterns, which can be easily detected and filtered by certain defense mechanisms such as norm clipping, trigger inversion and etc. In this work, we propose a novel generator-assisted backdoor attack, FTA, against FL defenses. We for the first time consider the natural stealthiness of triggers during global inference. In this method, we build a generative trigger function that can learn to manipulate the benign samples with naturally imperceptible trigger patterns (*stealthy*) and simultaneously make poisoned samples include similar hidden features of the attacker-chosen label. Moreover, our trigger generator repeatedly produces triggers for each sample (*flexibility*) in each FL iteration (*adaptivity*), allowing it to adjust to changes of hidden features between global models of different rounds. Instead of using universal and predefined triggers of existing works, we break this wall by providing three desiderate (i.e., stealthy, flexibility and adaptivity), which helps our attack avoid the presence of backdoor-related feature representations. Extensive experiments confirmed the effectiveness (above 98% attack success rate) and stealthiness of our attack compared to prior attacks on decentralized learning frameworks with eight well-studied defenses.

## 1 Introduction

Federated learning (FL) has recently provided practical performance in various real-world applications and tasks, such as prediction of oxygen requirements of symptomatic patients with COVID-19 (Dayan et al., 2021), autonomous driving (Nguyen et al., 2022a), Gboard (Yang et al., 2018) and Siri (Paulik et al., 2021). It supports collaborative training of an accurate global model by allowing multiple agents to upload local updates, such as gradients or weights, to a server without compromising local datasets. However, this decentralized paradigm unfortunately exposes FL to a security threat — backdoor attacks (Bhagoji et al., 2019; Xie et al., 2019; Wang et al., 2020; Zhang et al., 2022b; Li et al., 2023). Existing backdoor defenses on FL possess the capability to scrutinize the anomaly of malicious model updates. Prior attacks fail to achieve adequate stealthiness under those robust FL systems due to malicious parameter perturbations introduced by the backdoor task.

We summarize the following open problems from the existing backdoor attacks against FL[1]:

**P1: The abnormality of feature extraction in convolutional layers.** Existing attacks use patch-based triggers ("squares", "stripe" and etc.) (Bagdasaryan et al., 2020; Xie et al., 2019; Zhang et al., 2022b; Li et al., 2023) on a fixed position or semantic backdoor triggers (shared attributes within the same class) (Bagdasaryan et al., 2020; Wang et al., 2020). Consequently, the poisoned samples are misclassified by the victim model towards the target label after backdoor training. However, we found that prior attacks manipulate the samples with universal patterns along the whole training iterations, which fails to provide enough "stealthiness" of the hidden features of the poisoned samples. The backdoor training with such triggers attaches extra hidden features to the backdoor patterns or revises current hidden features from the feature space in benign classes domain. This makes the latent representations of poisoned samples extracted from filters *standalone* compared to the benign counterparts. **Figure 5 (a)** intuitively illustrates the statement. Therefore, unrestricted trigger patterns can cause aberrant weight changes in the filters for backdoor patterns. This abnormality induces

---

[1]Due to space limit, we review prior backdoor attacks and defenses on FL in Appendix A.1

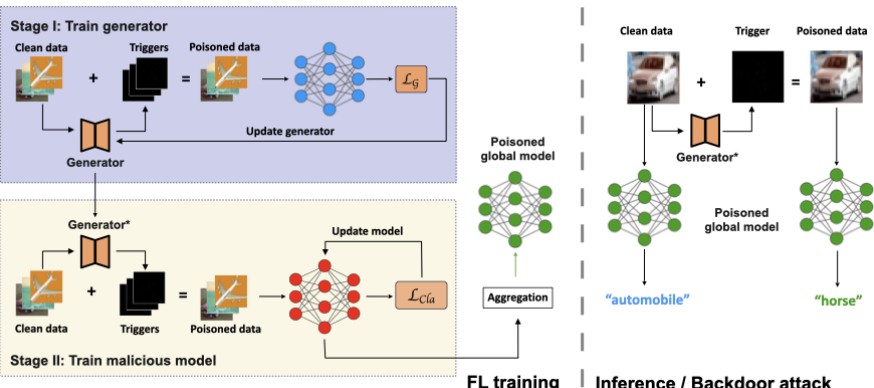

Figure 1: Overview of FTA. (I) Learn the optimal trigger generator $g_\xi$. (II) Train malicious model $f_\theta$. Inference/Backdoor Attack: The global model performs well on benign tasks while misclassifying the poisoned samples to the target label.

weight outliers which makes the backdoor attacks vulnerable to filter-wise adversarial pruning (Wu & Wang, 2021; Liu et al., 2018; Wu et al., 2020a).

**P2: The abnormality of backdoor routing in fully connected layers.** Compared with the benign model, the malicious model needs to be trained on one more task, i.e. backdoor task. Specifically, in fully connected (FC) layers, the backdoor task is to establish a *new* routing (Wang et al., 2018; Carnerero-Cano et al., 2023), separated from benign ones, between the independent hidden features of attacker's universal pattern and its corresponding target label, which yields an anomaly at the parameter level. The cause of this anomaly is natural, since the output neurons for the target label must contribute to both benign and backdoor routing, which requires significant weight/bias adjustments to the neurons involved. We note that last FC layer in the current mainstream neural networks are always with a large fraction of the total number of parameters (e.g., 98% for Classic CNN, 62% in ResNet18). As mentioned in (Rieger et al., 2022), the final FC layer of the malicious classifier presents significantly greater abnormality than other FC layers, with backdoor routing being seen as the secondary source of these abnormalities. Note that these abnormalities (**P1-2**) would arise in existing universal trigger designs under FL.

**P3: The perceptible trigger for inference.** Perhaps, it is not necessary to guarantee natural stealthiness of triggers on training data against FL, since its accessibility is limited to each client exclusively due to the privacy issue. However, we pay attention to the trigger stealthiness during the inference stage, in which a poisoned sample with a naturally stealthy trigger can mislead human inspection. The test input with perceptible perturbation in FL (Bagdasaryan et al., 2020; Xie et al., 2019; Zhang et al., 2022b; Li et al., 2023) can be easily identified by an evaluator or a user who can distinguish the difference between 'just' an incorrect classification/prediction of the model and the purposeful wrong decision due to a backdoor in the test/use stage.

**P1-3** can fatally harm the stealthiness and effectiveness of prior attacks under robust FL systems. The stealthiness issue can be seen in two aspects (trigger/routing). For **P3**, the visible fixed triggers contain independent hidden features, and these hidden features lead to a new backdoor routing as discussed in **P1-2**. Meanwhile, the backdoor inference stage cannot perform properly because those triggers are not sufficiently hidden. For example, we recall that DBA (Xie et al., 2019), Neurotoxin (Zhang et al., 2022b) and 3DFed (Li et al., 2023) use universal patterns that can be clearly filtered out by trigger inversion method such as FLIP (Zhang et al., 2023). Moreover, **P1-2** can cause weight dissimilarity between benign and backdoor routing. And this dissimilarity can be easily detected by cluster-based filtering, such as FLAME (Nguyen et al., 2022b). Efficiency problem is also striking for **P1-2** since extra computational budget is required to learn the new features of the poisoned data and to form the correspondent backdoor routing. In this work, we regard the problems **P1-3** as the *stealthiness* of backdoor attacks in the context of FL.

A natural question then arises: *could we eliminate the anomalies introduced by new backdoor features and routing (**i.e., tackling P1-2**) while making the trigger sufficiently stealthy for inference on decentralized scenario (**i.e., addressing P3**)?*

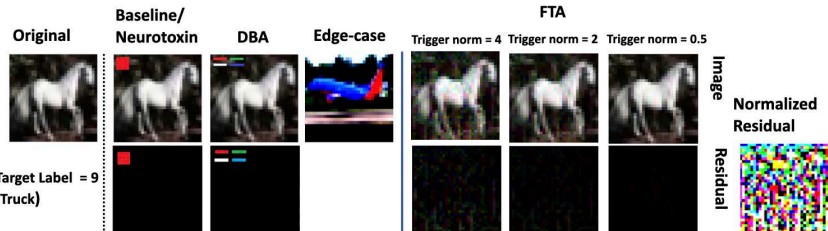

Figure 2: Visualization of backdoored images. **Top**: the original image; backdoored samples generated by baseline/Neurotoxin, DBA, Edge-case, and FTA; **Bottom**: the residual maps. Our flexible triggers appear as imperceptible noise.

To provide a concrete answer, we propose a stealthy generator-assisted backdoor attack, FTA, to adaptively (per FL iteration) provide triggers in a flexible manner (per sample) on decentralized setup. FTA achieves a satisfied stealthiness by producing imperceptible triggers with a generative neural network (GAN) (Goodfellow et al., 2014; Arjovsky et al., 2017) in a *flexible* way for each sample and in an *adaptive* manner during entire FL iterations. To address **P3**, our triggers should provide natural stealthiness to avoid inspection during inference. To solve **P1**, the difference of hidden features between poisoned data and benign counterparts should be minimized. Due to the imperceptibility between poisoned and benign data in latent representation, the correspondent backdoor routing will not be formed and thus **P2** is effectively addressed.

Specifically, the generator is learnt to produce triggers for each sample, which can ensure similar latent features of poisoned samples to benign ones with target label (**P1**). This idea can reduce the abnormality of creating an extra routing for backdoor in **P2** since the latent features make poisoned data "look like" benign ones with target label. Thus our trigger is less perceptible and more flexible than predefined patch-based ones in prior attacks (**P3**). Further, to make the flexible trigger robust and adaptive to the changes in global model, the generator is continuously trained across FL iterations. Compared with existing works using fixed and universal trigger patterns, we break this wall and for the first time make the generated trigger to be stealthy, flexible and adaptive in FL setups. Compared to universal trigger-based attacks, e.g., 3DFed, our trigger-assisted attack can effectively evade (universal) trigger inversion defense such as FLIP. Since our trigger generation method forces poisoned samples to share similar hidden feature as benign one, the benign routing can be mostly reused by poisoned data and thus the backdoor task is not purely learned from scratch. Our trigger generation ensures that poisoned samples have similar hidden features to benign ones, allowing poisoned data to reuse the benign routing. As a result, the backdoor task does not need to be learned entirely from scratch, thereby achieving high attack efficiency, as shown in Figure 3. Finally, we formulate the process of finding optimal trigger generator and training malicious model in a bi-level, non-convex and constrained optimization problem, and achieve optimum by proposing a simple but practical optimization process. We illustrate learning the trigger generator, training the malicious model and testing the backdoor in Figure 1, and showcase various backdoor images in Figure 2 to demonstrate the imperceptible perturbation by our generator.

Our main **contributions** are summarized as follows:

• We propose a stealthy generator-assisted backdoor attack (FTA) against robust FL. Instead of utilizing an universal trigger pattern, we design a novel trigger generator that produces naturally imperceptible triggers during inference stage. Our flexible triggers provide hidden feature similarity of benign data and successfully lead poisoned data to reuse benign routing of target label. Hereby FTA can avoid anomaly in parameter space and improve attack effectiveness.
• We design a new learnable and adaptive generator that can learn the flexible triggers for global model at current FL iteration to achieve the best attack effectiveness. We propose a bi-level and constrained optimization problem to find our optimal generator each iteration efficiently. We then formulate a customized learning process and solve it with reasonable complexity, making it applicable to the FL scenario.
• Finally, we present intensive experiments to empirically demonstrate that the proposed attack provides state-of-the-art effectiveness and stealthiness against existing eight well-study defense mechanisms under four benchmark datasets.

## 2 THREAT MODEL AND INTUITION

### 2.1 THREAT MODEL

**Attacker's Knowledge & Capabilities:** We consider the same threat model as in prior works (Bagdasaryan et al., 2020; Bhagoji et al., 2019; Wang et al., 2020; Zhang et al., 2022b; Shejwalkar et al., 2021; Panda et al., 2022), where the attacker can have full access to malicious agent device(s), local training processes and training datasets. Furthermore, we do not require the attacker to know the FL aggregation rules applied in the server.

**Attacker's Goal**: Unlike untargeted poisoning attacks (Jagielski et al., 2018) preventing the convergence of the global model, the goal of our attack is to manipulate malicious agents' local training processes to achieve high accuracy in the backdoor task without undermining benign accuracy.

### 2.2 OUR INTUITION

Recall that prior attacks use universal predefined patterns (see Figure 2) which cannot guarantee stealthiness (**P1-3**) since the poisoned samples are visually inconsistent with natural inputs. These universal triggers (including tail data) used in whole FL iterations with noticeable modification can introduce new hidden features during extraction and further influence the process of backdoor routing. Consequently, this makes prior attacks be easily detected by current robust defenses due to **P1-2**. Also, the inconsistency between benign and poisoned samples tends to be detected by defenders during the global inference (**P3**) and the triggers can be inversed in decentralized setup.

Compared to prior attacks that focus on manipulating parameters, we bridge the gap and focus on designing stealthy triggers. To address **P1-3**, a well-designed trigger should provide 4 superiorities: *i*) the poisoned sample is naturally stealthy to the original benign sample; *ii*) the trigger is able to achieve hidden feature similarity between poisoned and benign samples of target label; *iii*) the trigger can eliminate the anomaly between backdoor and benign routing during learning; *iv*) the trigger design framework can evade robust FL defenses. A practical and effective solution that provides these advantages over prior works simultaneously is the design of *flexible* triggers. The optimal flexible triggers are learnt to make latent representations of poisoned samples similar to benign ones, thereby making the reuse of benign routing possible, which can naturally diminish the presence of outlier at parameter level. We propose a learnable and adaptive trigger generator to produce flexible and stealthy triggers.

**v.s. Trigger generators in centralized setting.** One may argue that the attacker can simply apply a similar (trigger) generator in centralized setup (Doan et al., 2021b;a; Zhao et al., 2022b; Li et al., 2021b; Zhong et al., 2022) on FL to achieve imperceptible trigger and stealthy model update.
• **Stealthiness.** For example, the attacker can use a generator to produce imperceptible triggers for poisoned samples and make their hidden features similar to original benign samples' as in (Zhao et al., 2022b; Zhong et al., 2022). This, however, cannot ensure the indistinguishable perturbation of model parameters (caused by backdoor routing) during malicious training and fail to capture the stealthiness (in **P1-2**) in FL setup. This is so because it only constrains the distinction of the input domain and the hidden features between poisoned and benign samples other than the hidden features between poisoned and benign samples of **target** label. In other words, a centralized generator masks triggers in the input domain and feature space of benign samples, conceals the poisoned sample for visibility and feature representation, whereas this cannot ensure the absence of backdoor routing for poisoned data. A stealthy backdoor attack on FL should mitigate the routing introduced by backdoor task and guarantee the stealthiness of model parameters instead of just the hidden features of poisoned samples compared to their original inputs.
**Learning.** The centralized learning process of existing trigger generators cannot directly apply to decentralized setups due to the continuously changing of global model and time consumption of training trigger generator. As an example, IBA (Zhong et al., 2022) directly constrains the distance of feature representation between benign and poisoned samples. This approach cannot achieve satisfied attack effectiveness due to inaccurate hidden features of benign samples before global model convergence. In contrast, we propose a customized optimization method for FL scenarios that can learn the optimal trigger generator for global model of current iteration to achieve the best attack effectiveness and practical computational cost as depicted in Section 3.3 and Appendix A.10.
• **Defenses.** We note that the robust FL aggregator can only access local updates of all agents other

than local training datasets. The centralized backdoor attack does not require consideration of the magnitude of the malicious parameters. However, in reality, the magnitude of malicious updates is usually larger than that of benign updates under FL setups. In that regard, norm clipping can effectively weaken and even eliminate the impact of the backdoor (Sun et al., 2019; Shejwalkar et al., 2021). Based on the flexibility of our triggers, we advance the state-of-the-art by enhancing the stealthiness and effectiveness of the backdoor attack even against well-studied defenses such as trigger inversion method on FL, e.g. FLIP. We note that FLIP is effective in removing backdoors with patch-based triggers, whereas our proposed attack can effectively evade this SOTA defense.

# 3 PROPOSED METHODOLOGY: FTA

## 3.1 PROBLEM FORMULATION

Based on the federated scenario in Appendix A.1.1, the attacker $m$ trains the malicious models to alter the behavior of the global model $\theta$ under ERM as follows: $\theta_m^* = \arg\min_{\theta} \sum_{(x,y) \in D^{cln} \cup D^{bd}} \mathcal{L}(f_\theta(x), y)$, where $D^{cln}$ is clean training set and $D^{bd}$ is a small fraction of clean samples in $D^{cln}$ to produce poisoned data by the attacker. Each clean sample $(x, y)$ in the selected subset is transformed into a poisoned sample as $(\mathcal{T}(x), \eta(y))$, where $\mathcal{T} : \mathcal{X} \to \mathcal{X}$ is the trigger function and $\eta$ is the target labeling function. And the poison fraction is defined as $|D^{bd}|/|D^{cln}|$. During inference, for a clean input $x$ and its true label $y$, the learned $f$ behaves as: $f(x) = y, f(\mathcal{T}(x)) = \eta(y)$.

To generate a stealthy backdoor, our main goal is to learn a stealthy trigger function $\mathcal{T} : \mathcal{X} \to \mathcal{X}$ to craft poisoned samples and a malicious backdoor model $f_{\theta_m^*}$ to inject backdoor behavior into the global model with the followings: 1) the poisoned sample $\mathcal{T}(x)$ provides an imperceptible perturbation to ensure that we do not bring distribution divergences between clean and backdoor datasets; 2) the injected global classifier simultaneously performs indifferently on test input $x$ compared to its vanilla version but changes its prediction on the poisoned image $\mathcal{T}(x)$ to the target class $\eta(y)$; 3) the latent representation of backdoor sample $\mathcal{T}(x)$ is similar to its benign input $x$. Inspired by recent works in learning trigger function backdoor attacks (Cheng et al., 2021; Doan et al., 2021b; Nguyen & Tran, 2020; Zhao et al., 2022b), we propose to jointly learn $\mathcal{T}(\cdot)$ and poison $f_\theta$ via the following constrained optimization:

$$\min_{\theta} \sum_{(x,y) \in D^{cln}} \mathcal{L}(f_\theta^t(x), y) + \sum_{(x,y) \in D^{bd}} \mathcal{L}(f_\theta^t(\mathcal{T}_{\xi^*(\theta)}(x)), \eta(y))$$

$$s.t. \quad (i) \quad \xi^* = \arg\min_{\xi} \sum_{(x,y) \in D^{bd}} \mathcal{L}(f_\theta^t(\mathcal{T}_\xi(x)), \eta(y)) \tag{1}$$

$$(ii) \quad d(\mathcal{T}_\xi(x), x) \leq \epsilon$$

where $t$ is FL round, $d$ is a distance measurement function, $\epsilon$ is a constant threshold value to ensure a small perturbation by $l_2$-norm constraint, $\xi$ is the parameters of trigger function $\mathcal{T}(\cdot)$. In the above bilevel problem, we optimize a generative trigger function $\mathcal{T}_{\xi^*}$ that is associated with an optimally malicious classifier. The poisoning training finds the optimal parameters $\theta$ of the malicious classifier to minimize the linear combination of the benign and backdoor objectives. Meanwhile, the generative trigger function is trained to manipulate poisoned samples with imperceptible perturbation, while also finding the optimal trigger that can cause misclassification to the target label. The optimization in Equation (1) is a challenging task in FL scenario since the target classification model $f_\theta$ varies in each iteration and its non-linear constraint. Thus, the learned trigger function $\mathcal{T}_\xi$ is unstable based on dynamic $f_\theta$. For the optimization, we consider two steps: learning trigger generator and poisoning training, and further execute these steps respectively (not alternately) to optimize $f_\theta$ and $\mathcal{T}_\xi$. The details are depicted in Algorithm 1 (please see Appendix A.2 for more optimization details).

## 3.2 FTA TRIGGER FUNCTION

We train $\mathcal{T}_\xi$ based on a given generative network $g_\xi$, i.e., our FTA trigger generator. Similar to the philosophy of generative trigger technology (Doan et al., 2021b; Zhao et al., 2022b), we design our

trigger function to guarantee: 1) The perturbation of poisoned sample is imperceptible; 2) The trigger generator can learn features of input domain of target label to fool the global model. Given a benign sample $x$ and corresponding label $y$, we formally model $\mathcal{T}_\xi$ with restricted perturbation as follows:

$$\mathcal{T}_\xi(x) = x + g_\xi(x), \quad \|g_\xi(x)\|_2 \le \epsilon \quad \forall x, \quad \eta(y) = c, \tag{2}$$

where $\xi$ is the learnable parameters of the FTA trigger generator and $\epsilon$ is the trigger norm bound to constrain the value of the generative trigger norm. We use the same neural network architecture as (Doan et al., 2021b) to build our trigger generator $g_\xi$, i.e., an autoencoder or more complex U-Net structure (Ronneberger et al., 2015). The $l_2$-norm of the imperceptible trigger noise generated by $g_\xi$ is strictly limited within $\epsilon$ by: $\frac{g_\xi(x)}{max(1, \|g_\xi(x)\|_2/\epsilon)}$. Note that, under Equation (2), the distance $d$ in Equation (1) is $l_2$-norm on the image-pixel space between $\mathcal{T}_\xi(x)$ and $x$.

### 3.3 FTA's Optimization

To address the constrained optimization in Equation (1), existing function based attacks alternately updating $f_\theta$ while keeping $\mathcal{T}_\xi$ unchanged, or the other way round, for many iterations. However, according to our trials, we find that simply updating parameters makes the training process unstable and harms the backdoor performance. Inspired by (Doan et al., 2022), one local round of FTA attack is divided into two phases, and each phase is executed for only one iteration with fewer epochs. In phase one, we fix the classification model $f_\theta$ and only learn the trigger function $\mathcal{T}_\xi$. In phase two, we use the pre-trained $\mathcal{T}_{\xi^*}$ to generate the poisoned dataset and train the malicious classifier $f_\theta$. Since the number of poisoning epochs of malicious agents is fairly small, which means $f_\theta$ would not vary too much during poisoning training process, the hidden features of samples in target label extracted from $f_\theta$ will also remain similarly. The pre-trained $\mathcal{T}_{\xi^*}$ can still match with the final locally trained $f_\theta$.

In order to make flexible triggers generated by $g_\xi$ adaptive to global models in different rounds, $g_\xi$ should be continuously trained. If a malicious agent is selected more than one round to participate in FL iterations, it can keep training on previous pre-trained $g_\xi$ under new global model to make our flexible triggers match with hidden features of benign samples with target label from new model.

---

**Algorithm 1** FTA Backdoor Attack

---

**Input**: Clean dataset $D_{cln}$, Global model $f_\theta^t$ at round $t$, Learning rate of malicious model $\gamma_f$ and trigger function $\gamma_\mathcal{T}$, Batch of clean dataset $B_{cln}$ and poisoned dataset $B_{bd}$, Epochs to train trigger function $e_\mathcal{T}$ and malicious model $e_f$.
**Output**: Malicious model update $\delta^*$.

1: Initialize parameters of trigger function $\xi$ and global model: $f_\theta^t$.
2: Sample subset $D_{bd}$ from $D_{cln}$.
3: **// Stage I: Update flexible $\mathcal{T}$.**
4: Sample minibatch $(x, y) \in B_{bd}$ from $D_{bd}$
5: **for** $i = 1, 2, \cdots, e_\mathcal{T}$ **do**

6:     Optimize $\xi$ by using SGD with fixed $f_\theta^t$ on $B_{bd}$: $\xi \leftarrow \xi - \gamma_\mathcal{T} \overline{\nabla}_\xi \mathcal{L}(f_\theta^t(\mathcal{T}_\xi(x)), \eta(y))$
7: **end for**
8: $\xi^* \leftarrow \xi$
9: **// Stage II: Train malicious model $f^t$.**
10: Sample minibatch $(x, y) \in B_{cln}$ from $D_{cln}$ and $(x_m, y_m) \in B_{bd}$ from $D_{bd}$
11: **for** $i = 1, 2, \cdots, e_f$ **do**
12:     Optimize $\theta$ by using SGD with fixed $\mathcal{T}_{\xi^*}$ on $B_{cln}$, $B_{bd}$: $\theta \leftarrow \theta - \gamma_f \nabla_\theta(\mathcal{L}(f_\theta^t(x, y)) + \mathcal{L}(f_\theta^t(\mathcal{T}_\xi(x_m)), \eta(y_m)))$
13: **end for**
14: $\theta^* \leftarrow \theta$
15: Compute malicious update: $\delta^* \leftarrow \theta^* - \theta$

---

## 4 ATTACK EVALUATION

### 4.1 EXPERIMENTAL SETUP

**Datasets and Models.** We demonstrate the effectiveness of FTA backdoor through comprehensive experiments on four publicly available datasets, namely Fashion-MNIST (Xiao et al., 2017), FEM-NIST (Caldas et al., 2018), CIFAR-10 (Krizhevsky et al., 2009), and Tiny-ImageNet (Le & Yang, 2015). The classification model used in the experiments includes Classic CNN, VGG11 (Simonyan & Zisserman, 2015), and ResNet18 (He et al., 2016). These datasets and models are representative

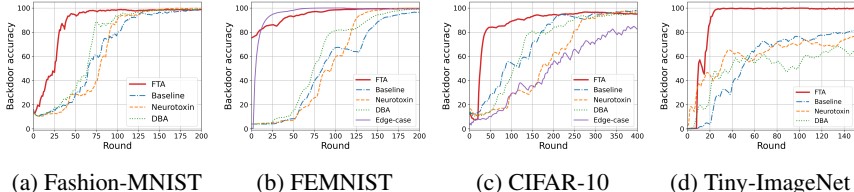

| (a) Fashion-MNIST | (b) FEMNIST | (c) CIFAR-10 | (d) Tiny-ImageNet |

Figure 3: Fixed-frequency attack performance under `FedAvg`. FTA is more effective than others.

and commonly used in existing backdoor and FL research works. The overview of our models is described in Appendix A.6. The details of tasks are depicted in Appendix A.3.

**Attack Settings.** As in Neurotoxin (Zhang et al., 2022b), we assume that the attacker can only compromise a limited number of agents (<1% ) in practice (Shejwalkar et al., 2021) and uses them to launch the attack by uploading manipulated gradients to the server. Malicious agents can only participate in a constrained number of training rounds in FL settings. Note even if the attacker has above restrictions, our attack can still be effective, stealthy and robust against defenses (see Figures 3 and 4). Also, the attack effectiveness should last even though the attacker stops the attack under robust FL aggregators (see Figure 6 in Appendix A.4.1). We test stealthiness and durability of FTA with two attack modes respectively, i.e., fixed-frequency and few-shot as Neurotoxin. *i*) Fixed-frequency mode: The server randomly chooses 10 agents among all agents. The attacker controls exactly one agent in each round in which they participate. For other rounds, 10 benign agents are randomly chosen among all agents. *ii*) Few-shot mode: The attacker participates only in `Attack_num` rounds. During these rounds, we ensure that one malicious agent is selected for training. After Attack_num rounds or backdoor accuracy has reached 95%, the attack will stop. Under this setting, the attack can take effect quickly, and gradually weaken by benign updates after the attack is stopped.

**Evaluation Metrics.** We evaluate the performance based on backdoor accuracy (BA) and benign accuracy according to the following criteria: effectiveness and stealthiness against current SOTA defense methods under fixed-frequency mode, durability evaluated under few-shot mode.

**Comparison.** We compare FTA with three SOTA attacks, namely DBA, Neurotoxin and Edge-case (Wang et al., 2020), and the baseline attack method described in (Zhang et al., 2022b) under different settings and ten defenses (a variant of norm clipping based on (Sun et al., 2019), FLAME, Multi-Krum (Blanchard et al., 2017), Trimmed-mean (Yin et al., 2018), RFA (Pillutla et al., 2022), SparseFed (Panda et al., 2022), SignSGD (Bernstein et al., 2019), Foolsgold (Fung et al., 2020)), RLR (Ozdayi et al., 2021) and Pruning (Wu et al., 2020b). We show that FTA outperforms 3 SOTA attacks (under 10 robust FL defenses) by conducting experiments on different computer vision tasks (please see Appendix A.7 for more experimental setup details).

## 4.2 ATTACK EFFECTIVENESS

**Attack effectiveness under fixed-frequency mode.** Compared to the attacks with unified triggers, FTA converges much faster and delivers the best BA in all cases since our poisoned data can reuse benign routing of target label, see Figure 3. It can yield a high backdoor accuracy on the server model within very few rounds (<50) and maintain above 97% accuracy on average. Especially in Tiny-ImageNet, FTA reaches 100% accuracy extremely fast, with at least 25% advantage compared to others. In CIFAR-10, FTA achieves nearly 83% BA after 50 rounds which is 60% higher than other attacks on average. There is only <5% BA gap between FTA and Edge-case on FEMNIST in the beginning and later, they reach the same BA after 100 rounds. We note that the backdoor task of Edge-case in FEMNIST is relatively easy, mapping 7-like images to the target label of digit "1", which makes its convergence slightly faster than ours.

**Attack effectiveness under few-shot mode.** As an independent interest, we test the durability of the attacks during training stage in this setting. According to Appendix A.4.1, FTA has long-term attack effectiveness even if we stop attacking early since the poisoned data with our well-learned triggers contain similar features to benign data and can be naturally misclassified into the target label with certain confidence by server model. Due to space limit, please see Appendix A.4.1 for more details.

**Influence on Benign accuracy and computational cost.** We include all benign accuracy results across tasks in Appendix A.5. Like other SOTA attacks, FTA has a minor effect (no more than 1.5%)

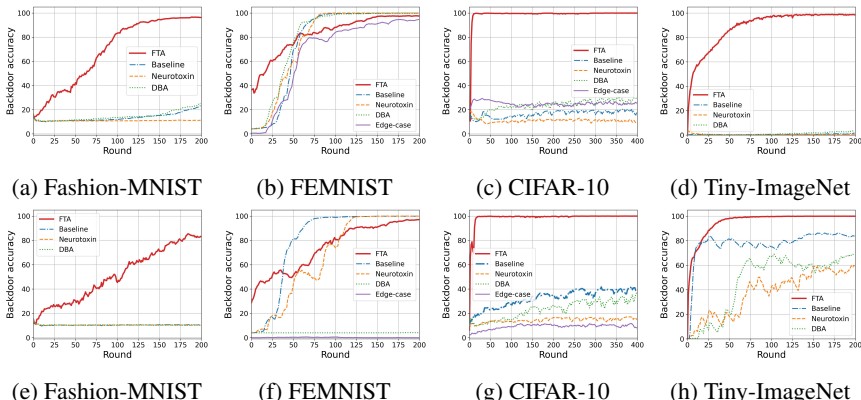

(a) Fashion-MNIST      (b) FEMNIST      (c) CIFAR-10      (d) Tiny-ImageNet

(e) Fashion-MNIST      (f) FEMNIST      (g) CIFAR-10      (h) Tiny-ImageNet

Figure 4: Attack stealthiness against defenses. (a)-(d): The variant of norm clipping; (e)-(h): FLAME.

on benign accuracy. Our attack does not significantly increase the computational and time cost due to our optimization procedure (see Appendix A.10 for details).

## 4.3 STEALTHINESS AGAINST DEFENSIVE MEASURES

We test the stealthiness **(P1-2)** and robustness of FTA and other attacks using 8 SOTA robust FL defenses introduced in Appendix A.1.3, such as norm clipping and FLAME, under fixed-frequency scenarios. All four tasks are involved in this defense evaluation. The results, see Figure 4 show that FTA can break the listed defenses. Beyond this, we also evaluate different tasks on Multi-Krum, Trimmed-mean, RFA, SignSGD, Foolsgold and SparseFed. FTA maintains its stealthiness and robustness under these defenses. We put results of compared attacks under defenses in Appendix A.4.

### 4.3.1 RESISTANCE TO VECTOR-WISE SCALING

We use the norm clipping as the vector-wise scaling defense method, which is regarded as a potent defense and has proven effective in mitigating prior attacks (Shejwalkar et al., 2021). On the server side, norm clipping is applied on all updates before performing `FedAvg`. Inspired by (Sun et al., 2019), we utilize the variant of this method in our experiments. As introduced in Appendix A.3, if we begin the attack from scratch, the norm of benign updates will be unstable and keep fluctuating, making us hard to set a fixed norm bound for all updates. We here filter out the biggest and smallest updates and compute the average norm magnitude based on the rest updates, and set it as the norm bound in current FL iteration.

As shown in Figure 4 (a)-(d), this variant of norm clipping can effectively undermine prior attacks in Fashion-MNIST, CIFAR-10, and Tiny-ImageNet. It fails in FEMNIST because benign updates have a larger norm (for example, 1.2 in FEMNIST at round 10, but only 0.3 in Fashion-MNIST), which cannot effectively clip the norm of malicious updates, thus resulting in a higher BA of existing attacks. We see that FTA provides the best BA which is less influenced by clipping than others. FTA only needs a much smaller norm to effectively fool the global model. Although converging a bit slowly in FEMNIST, FTA can finally output a similar performance (above 98%) compared to others.

### 4.3.2 RESISTANCE TO CLUSTER-BASED FILTERING

The cluster-based filtering defense method is FLAME, which has demonstrated its effectiveness in mitigating SOTA attacks against FL. It mainly uses HDBSCAN clustering algorithm based on cosine similarity between all updates and strains the updates with the least similarity compared with other updates. In Figure 4 (e)-(h), we see that FLAME can effectively sieve malicious updates of other attacks in Fashion-MNIST and CIFAR-10, but provides relatively weak effectiveness in FEMNIST and Tiny-ImageNet. This is so because data distribution among different agents are fairly in non-i.i.d. manner. Cosine similarity between benign updates is naturally low, making malicious update possibly evade from the clustering filter.

Similar to result of Multi-Krum (see Appendix A.4.2), FTA achieves >99% BA and finishes the convergence within 50 rounds in CIFAR-10 and Tiny-ImageNet, while delivering an acceptable

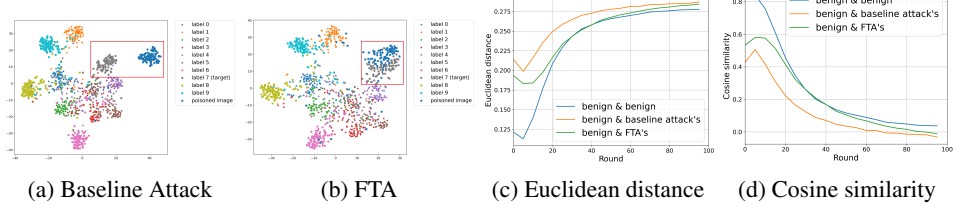

|  (a) Baseline Attack | (b) FTA | (c) Euclidean distance | (d) Cosine similarity |

Figure 5: (a)-(b): T-SNE visualization of hidden features of input samples in CIFAR-10. The hidden features between poisoned and benign samples of target label is indistinguishable in FTA framework. (c)-(d): Similarity comparison between benign & malicious updates. FTA's malicious updates is more similar to benign updates than the baseline attack's.

degradation of accuracy, <20%, in Fashion-MNIST. In FEMNIST, FTA converges slightly slower than baseline and Neurotoxin but eventually maintains a similar accuracy with only 2% difference. The result proves that FTA enforces malicious updates to have highly cosine-similarity against benign updates due to same reason in Appendix A.4.2, so it can bypass defenses based on model similarity.

## 4.4 EXPLANATION VIA FEATURE VISUALIZATION BY T-SNE

We use t-SNE (Van der Maaten & Hinton, 2008) visualization result on CIFAR-10 to illustrate why FTA is more stealthy than the attacks without "flexible" triggers. We select 1,000 images from different classes uniformly and choose another 100 images randomly from the dataset and add triggers to them (in particular, patch-based trigger "square" in baseline method, flexible triggers in FTA). To analyze the hidden features of these samples, we use two global poisoned models injected by baseline attack and FTA respectively. We exploit the output of each sample in the last convolutional layer as the feature representation. Next, we apply dimensionality reduction techniques and cluster the latent representations of these samples using t-SNE. From Figure 5 (a)-(b), We see that in the baseline, the distance of clusters between images of the target label "7" and the poisoned images are clearly distinguishable. So the parameters responsible for backdoor routing should do adjustments to map the hidden representations of poisoned images to target label. In FTA, the hidden features of poisoned data overlapped with benign data of target label, which eliminates the anomaly in **feature extraction (P1)**. FTA can reuse the benign routing in FC layers for backdoor tasks, resulting in much less abnormality in **backdoor routing (P2)**, thus the malicious updates can be more similar to benign ones, see Figure 5 (c)-(d), producing a natural parameter stealthiness.

## 4.5 NATURAL STEALTHINESS

We evaluate natural stealthiness of our poisoned data by SSIM (Wang et al., 2004) and LPIPS (Zhang et al., 2018) to indicate that **P3** is well addressed by flexible triggers (see Appendix A.9 for results).

## 4.6 ABLATION STUDY IN FTA ATTACK

We here analyze several hyperparameters that are critical for the FTA's performance including trigger size, poison fraction and batch size of our generator (please see Appendix A.8 for details).

## 5 CONCLUSION

We design an effective and stealthy backdoor attack against FL called FTA by learning an adaptive generator to produce imperceptible and flexible triggers, making poisoned samples have similar hidden features to benign samples with target label. FTA can provide stealthiness and robustness in making hidden features of poisoned samples consistent with benign samples of target label; reducing the abnormality of parameters during backdoor task training; manipulating triggers with imperceptible perturbation for training/testing stage; learning the adaptive trigger generator across different FL rounds to generate flexible triggers with best performance. The empirical experiments demonstrate that FTA can achieve a practical performance to evade SOTA FL defenses. Due to the space limit, we present discussions on the proposed attack and experiments in Appendix A.12. We hope this work can inspire follow-up studies that provide more secure and robust FL aggregation algorithms.

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

# A  APPENDIX

## A.1  RELATED WORK

### A.1.1  FEDERATED LEARNING

Consider the empirical risk minimization (ERM) in FL setting where the goal is to learn a global classifier $f_\theta : \mathcal{X} \to \mathcal{Y}$ that maps an input $x \in \mathcal{X}$ to a target label $y \in \mathcal{Y}$. Recall that the FL server cannot access to training dataset. It aggregates the parameters/gradients from local agents performing centralized training with local datasets. The de-facto standard rule for aggregating the updates is so-called `FedAvg` (McMahan et al., 2017). The training task is to learn the global parameters $\theta$ by solving the finite-sum optimization: $\min_\theta f_\theta = \frac{1}{n} \sum_{i=1}^{n} f_{\theta_i}$, where $n$ is the number of participating agents. At round $t$, the server $S$ randomly selects $n^t \in \{1, 2, ..., n\}$ agents to participate in the aggregation and sends the global model $\theta^t$ to them. Each of the agents $i$ trains its local classifier $f_{\theta_i} : \mathcal{X}_i \to \mathcal{Y}_i$ with its local dataset $D_i = \{(x_j, y_j) : x_j \in \mathcal{X}_i, y_j \in \mathcal{Y}_i, j = 1, 2, ..., N\}$ for some epochs, where $N = |D_i|$, by certain optimization algorithm, e.g., stochastic gradient descent (SGD). The objective of agent $i$ is to train a local model as: $\theta_i^* = \underset{\theta^t}{\operatorname{argmin}} \sum_{(x_j, y_j) \in D_i} \mathcal{L}(f_{\theta^t}(x_j), y_j)$, where $\mathcal{L}$ stands for the classification loss, e.g., cross-entropy loss. Then agent $i$ computes its local update as $\delta_i^t = \theta_i^* - \theta^t$, and sends back to $S$. Finally, the server aggregates all updates and produces the new global model with an average $\theta^{t+1} = \theta^t + \frac{\gamma}{|n^t|} \sum_{i \in n^t} \delta_i^t$. where $\gamma$ is the global learning rate. When the global model $\theta$ converges or the training reaches a specific iteration upper bound, the aggregation process terminates and outputs a final global model. During inference, given a benign sample $x$ and its true label $y$, the learned global classifier $f_\theta$ will behave well as: $f_\theta(x) = y$.

Optimizations of FL have been proposed for various purposes, e.g., privacy (Bonawitz et al., 2016), security (Blanchard et al., 2017; Zhao et al., 2022a), heterogeneity (Li et al., 2020), communication efficiency (Liu et al., 2019; Jiang et al., 2020) and personalization issues (Li et al., 2021a; Yu et al., 2020).

### A.1.2  BACKDOOR ATTACKS

**Backdoor Attacks on FL.** Current backdoor attacks can poison data and models. In data poisoning (Shen et al., 2016; Xie et al., 2019), the attacker poisons the benign samples with a trigger pattern and marks them as a target label in order to induce the model to misbehave by training this poisoned dataset. As for model poisoning (Bagdasaryan et al., 2020; Wang et al., 2020), the attacker manipulates the training process by modifying parameters and scaling the malicious update to maximize the attack effectiveness while evading anomaly detection of robust FL aggregators (Blanchard et al., 2017; Sun et al., 2019; Panda et al., 2022; Nguyen et al., 2022b; Rieger et al., 2022; Yin et al., 2018; Bernstein et al., 2019).

The most well-known backdoor attack on FL is introduced in (Bagdasaryan et al., 2020), where the adversary scales up the weights of malicious model updates to maximize attack impact and replace the global model with its malicious local model. To fully exploit the distributed learning methodology of FL, the local trigger patterns are used in (Xie et al., 2019) to generate poisoned images for different malicious models, while the data from the tail of the input data distribution is leveraged in (Wang et al., 2020). Durable backdoor attacks are proposed in (Zhang et al., 2022b; Dai & Li, 2023), and make attack itself more persistent in the federated scenarios. We state that this kind of attacks mainly focuses on the persistence, whereas our focus is on stealthiness.

Existing works reply on a universal trigger or tail data, which do not fully exploit the "attribute" of trigger. Our design is fully applicable and complementary to prior attacks. By learning a stealthy trigger generator and injecting the sample-specific triggers, we can significantly decrease the anomalies in **P1-3** and reinforce the stealthiness of backdoor attacks.

**Function-based Backdoor Attacks.** Existing backdoor attacks can leverage generative functions to produce triggers. (Nguyen & Tran, 2020) implements a trigger generator to make triggers vary from input to input. To improve the invisibility of triggers, (Doan et al., 2021b) proposes a bi-level problem to learn a generator to produce invisible sample-specific triggers. Further, (Doan et al., 2021a) extends the concept of imperceptible backdoors from the input to feature space, which learns a trigger generator to constrain the similarity of hidden features between posioned and clean data. To

improve the stealthiness of triggers on latent representations, (Zhao et al., 2022b) adaptively learns the generator by constraining the latent layers, which makes triggers more invisible in both input and latent feature space. Moreover, (Doan et al., 2022) proposed a generator that can learn invisible triggers with arbitrary target class. Most works propose their customized bi-level optimization problems and solve them by alternatively learning their generative and classification models. There is less literature considering function-based attacks against FL, we thus propose a stealthy backdoor attack to eliminate the abnormality of parameters and bridge the gap between the centralized and federated scenarios.

### A.1.3 BACKDOOR DEFENSES ON FL

There are a number of defenses that provide empirical robustness against backdoor attacks.

**Dimension-wise filtering.** Trimmed-mean (Yin et al., 2018) aggregates each dimension of model updates of all agents independently. It sorts the parameters of the $j^{th}$-dimension of all updates and removes $m$ of the largest and smallest parameters in that dimension. Finally, it computes the arithmetic mean of the rest parameters as the aggregate of dimension $j$. Similarly, Median (Yin et al., 2018) takes the arithmetic median value of each dimension for aggregation. SignSGD (Bernstein et al., 2019) only aggregates the signs of the gradients (of all agents) and returns the sign to agents for updating the local models.

**Vector-wise scaling.** Norm clipping (Sun et al., 2019) bounds the $l_2$-norm of all updates to a fixed threshold due to high norms of malicious updates. For a threshold $\tau$ and an update $\nabla$, if the norm of the update $||\nabla|| > \tau$, $\nabla$ is scaled by $\frac{\tau}{||\nabla||}$. The server averages all the updates, scaled or not, for aggregation.

**Vector-wise filtering.** Krum (Blanchard et al., 2017) selects a local model, with the smallest Euclidean distance to $n - f - 1$ of other local models, as the global model. A variant of Krum called Multi-Krum (Blanchard et al., 2017) selects a local model using Krum and removes it from the remaining models repeatedly. The selected model is added to a selection $S$ until $S$ has $c$ models such that $n - c > 2m + 2$, where $n$ is the number of selected models and $m$ is the number of malicious models. Finally, Multi-Krum averages the selected model updates. RFA (Pillutla et al., 2022) aggregates model updates and makes FedAvg robust to outliers by replacing the averaging aggregation with an approximate geometric median.

**Certification.** CRFL (Xie et al., 2021) provides certified robustness in FL frameworks. It exploits parameter clipping and perturbing during federated averaging aggregation. In the test stage, it constructs a "smoothed" classifier using parameter smoothing. The robust accuracy of each test sample can be certified by this classifier when the number of compromised clients or perturbation to the test input is below a certified threshold.

**Sparsification.** SparseFed (Panda et al., 2022) performs norm clipping to all local updates and averages the updates as the aggregate. $Top_k$ values of the aggregation update are extracted and returned to each agent who locally updates the models using this sparse update.

**Cluster-based filtering.** Recently, (Nguyen et al., 2022b) proposed a defending framework FLAME based on the clustering algorithm (HDBSCAN) which can cluster dynamically all local updates based on their cosine distance into two groups separately. FLAME uses weight clipping for scaling-up malicious weights and noise addition for smoothing the boundary of clustering after filtering malicious updates. By using HDBSCAN, (Rieger et al., 2022) designed a robust FL aggregation rule called DeepSight. Their design leverages the distribution of labels for the output layer, output of random inputs, and cosine similarity of updates to cluster all agents' updates and further applies the clipping method.

### A.2 THE PROCEDURE OF FTA OPTIMIZATION

In case of collusion between more than one malicious agent device, the local datasets owned by these devices are in non-i.i.d. manner. Their local trigger generators $g_{\xi_i}$ are trained by these local datasets. This kind of dataset bias can degrade attack effectiveness since their malicious updates are for local triggers from different $g_{\xi_i}$ and cannot be merged together to yield a better attack performance. To resolve this problem, we develop a practical solution. Before starting the FTA backdoor attack, the

malicious agents can share a portion of their local datasets to form a universal poisoned dataset (for all the malicious agents), so that their local generators $g_{\xi_i}$ can produce the same triggers.

## A.3 DETAILS OF THE TASKS

The details of 4 computer vision tasks are described in Table 1. To prove the stealthiness and further robustness against defenses of FTA, we use a decentralized setting with non-i.i.d. data distribution among all agents. The attacker chooses the all-to-one type of backdoor attack (except Edge-case (Wang et al., 2020)), fooling the global model to misclassify the poisoned images of any label to an attacker-chosen target label. Following a practical scenario for the attacker given in (Zhang et al., 2022b), *10* agents among thousands of agents are selected for training in each round and their updates are used for aggregation and updating the server model. We apply backdoor attacks from different phases of training. In FEMNIST task, we follow the same setting as (Xie et al., 2019), where the attacker begins to attack when the benign accuracy of global models starts to converge. For other tasks, we perform backdoor attacks at the beginning of FL training. In this sense, as mentioned in (Xie et al., 2019), benign updates are more likely to share common patterns of gradients and have a larger magnitude than malicious updates, which can significantly restrict the effectiveness of malicious updates. Note we consider such a setting for the bottom performance of attacks and further, we still see that our attack performs more effectively than prior works in this case (see Figure 3).

Table 1: The datasets, and their corresponding models and hyperparameters.

| | Fahion-MNIST | FEMNIST | CIFAR-10 | Tiny-ImageNet |
|---|---|---|---|---|
| Classes | 62 | 10 | 10 | 200 |
| Size of training set | 60000 | 737837 | 50000 | 100000 |
| Size of testing set | 10000 | 80014 | 10000 | 10000 |
| Total agents | 2000 | 3000 | 1000 | 2000 |
| Malicious agents | 2 | 3 | 1 | 2 |
| Agents per FL round | 10 | 10 | 10 | 10 |
| Phase to start attack | Attack from scratch | Attack after convergence | Attack from scratch | Attack from scratch |
| Poison fraction | 0.2 | | | |
| Trigger size | 1 | 1.5 | 1.5 | 3 |
| Dataset size of trigger generator | 256 | | | |
| Epochs of benign task | 2 | 4 | 5 | 5 |
| Epochs of backdoor task | 5 (FTA: 2) | 10 (FTA: 4) | 10 (FTA: 5) | 10 (FTA: 5) |
| Learning rate of trigger generator | 0.01 | 0.01 | 0.001 | 0.01 |
| Epochs of trigger generator | 20 | 20 | 30 | 30 |
| Local data distribution | non-i.i.d. | | | |
| Classification model | Classic CNN | Classic CNN | ResNet-18 | ResNet-18 |
| Trigger generator model | Autoencoder | Autoencoder | U-Net | Autoencoder |
| Learning rate of benign task | 0.1 | 0.01 | 0.01 | 0.001 |
| Learning rate of backdoor task | 0.1 | 0.01 | 0.01 | 0.01 |
| Edge-case | FALSE | TRUE | TRUE | FALSE |
| Other hyperparameters | Momentum:0.9, Weight Decay: $10^{-4}$ | | | |

## A.4 FTA AGAINST OTHER DEFENSES

Besides the defense methods used in the main body, we test the performance of FTA under Multi-Krum, Trimmed-mean, RFA, SignSGD, Foolsgold and SparsedFed. The results prove that FTA is able to evade the defenses.

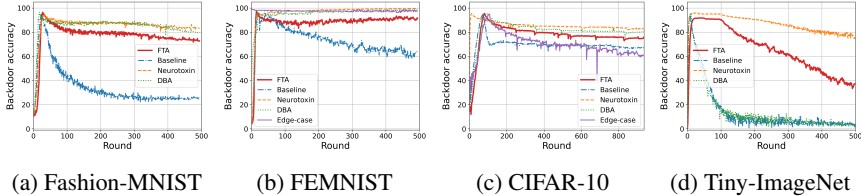

|   (a) Fashion-MNIST   |   (b) FEMNIST   |   (c) CIFAR-10   |   (d) Tiny-ImageNet   |

Figure 6: Few-shot attack performance under `FedAvg`. FTA is more durable than baseline.

### A.4.1 ATTACK EFFECTIVENESS UNDER FEW-SHOT MODE.

In our experiments, the Attack_num is 100 for all attacks, and the total FL round is 1000 for CIFAR-10, and 500 for other datasets. The results under few-shot settings are shown in Figure 6. All attacks reach a high BA rapidly after consistently poisoning the server model, then BA gradually drops after stopping attacking and the backdoor injected into the server model is gradually weakened by the aggregation of benign updates. FTA's performance drops much slower than the baseline attack. For example, in Fashion-MNIST and after 500 rounds, FTA still remains 73% BA, which is only 9% less than Neurotoxin, 61% higher than the baseline. Moreover, FTA can beat DBA and the baseline on Tiny-ImageNet. After 500 rounds, FTA maintains 37% accuracy while the baseline and DBA only have 5%, which is 45% less than Neurotoxin. However, Neurotoxin cannot provide the same stealthiness as shown in following comparison under robust FL defenses. Since malicious and benign updates have a similar direction by FTA, the effectiveness of FTA's backdoor can survive after few-shot attack. The results prove the durability of FTA.

### A.4.2 RESISTANCE TO VECTOR-WISE FILTERING

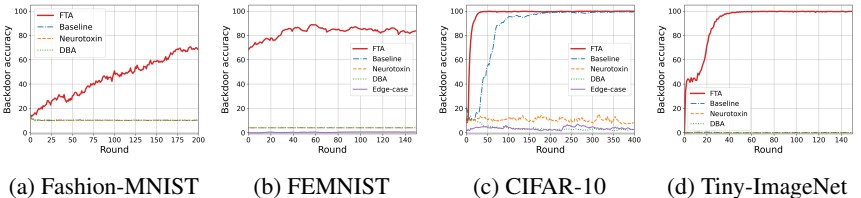

|   (a) Fashion-MNIST   |   (b) FEMNIST   |   (c) CIFAR-10   |   (d) Tiny-ImageNet   |

Figure 7: The effectiveness of attack under Multi-Krum in 4 tasks.

Multi-Krum is used as the vector-wise defense method. As described in Appendix A.1.3, it calculates the Euclidean distance between all updates and selects $n - f - 1$ updates with the smallest Euclidean distances for aggregation. In Figure 7, the defense manages to filter out almost all malicious updates of prior attacks and effectively degrade their attacks' performance. In contrast, local update of FTA cannot be easily filtered and thus FTA outperforms others. In CIFAR-10 and Tiny-ImageNet, the attack performance is steady for FTA (nearly 100%) within 40 rounds to converge. In FEMNIST, Multi-Krum only results in a 10% BA degradation for FTA while BAs of others are restricted to 0%. In Fashion-MNIST, Multi-Krum can sieve malicious updates of FTA occasionally, leading to a longer convergence time, but still fails to completely defend the FTA. Malicious updates produced by FTA (which successfully eliminates the anomalies in **P1-2**) are with a similar Euclidean distance compared to benign updates, making them more stealthy than other attacks'.

### A.4.3 RESISTANCE TO DIMENSION-WISE FILTERING

We choose Trimmed-mean as the representative of dimension-wise filtering. As mentioned in Appendix A.1.3, the dimensions of updates are sorted respectively, and the top $m$ highest and smallest updates are removed, and the arithmetic mean of the rest parameters is computed for aggregated updates. In our experiments, $m$ is set as 2 because we assume there is no more than one malicious agent during FL iteration, and setting a higher $m$ can result in lower convergence. As shown in Figure 8, Trimmed-mean successfully filters out the compared attacks in Fashion-MNIST and Tiny-ImageNet, and its effects are weakened in CIFAR-10 and FEMNIST. However, FTA survives in all

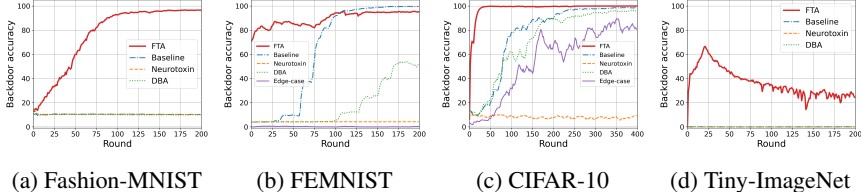

(a) Fashion-MNIST     (b) FEMNIST     (c) CIFAR-10     (d) Tiny-ImageNet

Figure 8: The effectiveness of attack under Trimmed-mean in 4 tasks.

four tasks and performs the best under trimmed-mean. In CIFAR-10, it completes the convergence within 30 rounds and remains 99.9% BA. In Fashion-MNIST and FEMNIST, FTA takes above 50 rounds to fully converge, and the final accuracy manages to reach 96%. The performance of FTA is significantly degraded in Tiny-ImageNet, but still with 30% advantage over other attacks on average. The update of FTA shares a similar weights/biases distribution of benign updates. This ensures our attack to defeat the defenses based on dimension-wise filtering.

### A.4.4 RESISTANCE TO RFA

In Figure 9, FTA provides the best performance among others in Fashion-MNIST, CIFAR-10 and Tiny-ImageNet. In FEMNIST, it converges much faster than prior attacks. Although its accuracy is 8% lower than the baseline in the middle of training, FTA achieves the same performance at the end (of training).

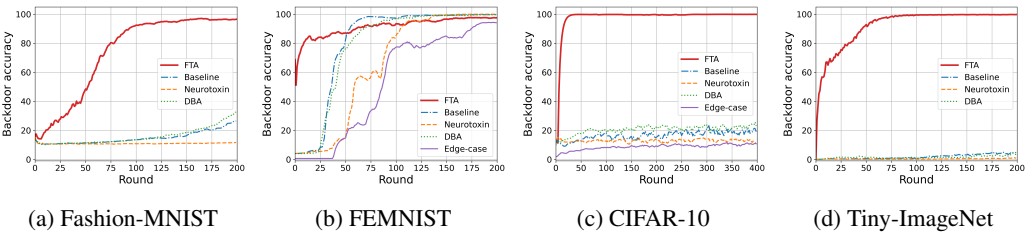

(a) Fashion-MNIST     (b) FEMNIST     (c) CIFAR-10     (d) Tiny-ImageNet

Figure 9: The effectiveness of attack under RFA in 4 tasks.

### A.4.5 RESISTANCE TO SIGNSGD AND RLR

As shown in Figure 10 (a)-(b), SignSGD mitigates prior backdoor attacks with a universal trigger pattern. However, FTA still defeats it and remains 94% and 99% BA on Fashion-MNIST and Tiny-ImageNet, respectively.

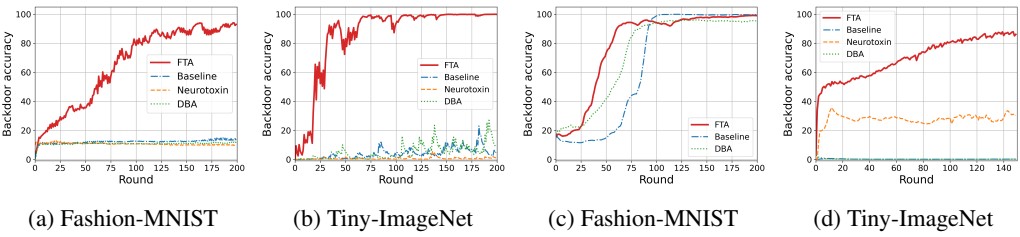

(a) Fashion-MNIST     (b) Tiny-ImageNet     (c) Fashion-MNIST     (d) Tiny-ImageNet

Figure 10: (a)-(b): The effectiveness of attack under SignSGD in Fashion-MNIST and Tiny-ImageNet. (c): The effectiveness of attack under Foolsgold in Fashion-MNIST. (d): The effectiveness of attack under SparseFed in Tiny-ImageNet.

RLR can effectively filter out malicious updates, we test FTA under RLR as the variant of SignSGD. As shown in Figure 11 (a)-(b), FTA achieves above 98% backdoor accuracy in both i.i.d and non-i.i.d manners.

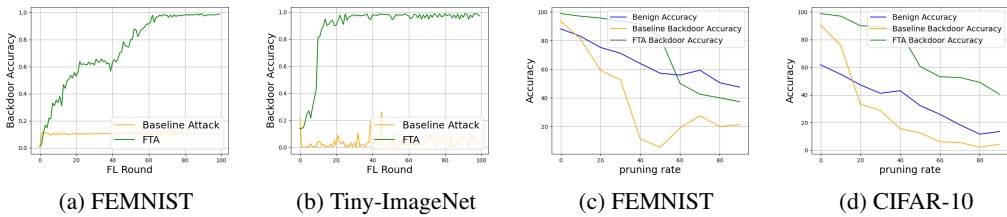

|     (a) FEMNIST     |    (b) Tiny-ImageNet    |     (c) FEMNIST     |     (d) CIFAR-10     |

Figure 11: (a)-(b): The effectiveness of attack under RLR in FEMNIST and Tiny-ImageNet. (c)-(d): The effectiveness of attack under Pruning in FEMNIST and CIFAR-10.

### A.4.6    RESISTANCE TO FOOLSGOLD

From Figure 10 (c), we see that Foolsgold hinders the convergence speed of FTA in Fashion-MNIST, which requires FTA to perform extra 25 rounds for convergence. In this sense, FTA still converges much faster than others.

### A.4.7    RESISTANCE TO SPARSIFICATION

We choose SparseFed as the representative of the sparsification defense. In Figure 10 (d), only Neurotoxin and FTA are capable of breaking through SparseFed on Tiny-ImageNet. The BA of Neurotoxin exhibits fluctuations (between 22% and 36%) throughout the training process, unable to maintain a continuous rise. In contrast, FTA demonstrates the ability to consistently poison the global model and later achieves an impressive accuracy of 90% by round 150. The reason for the above performance difference is that the backdoor task of FTA captures imperceptible perturbations on model parameters, which eliminates the anomalies of poisoning training. The backdoor tasks trained by FTA are more likely to contribute to the same dimensions of gradients as benign updates. Consequently, the top-$k$ filtering mechanism implemented in the server side is ineffective to filter out FTA's backdoor effect.

### A.4.8    RESISTANCE TO POST-TRAINING STAGE DEFENSE

We choose Pruning as a strong post-training stage defense. The attack effectiveness of FTA under Pruning is illustrated in Figure 11 (c)-(d). Figure 11 (c) demonstrates that even with a 50% prune ratio, FTA maintains 80% backdoor accuracy in FEMNIST. To reduce the backdoor accuracy to below 40%, it must prune 90% of neurons in CIFAR-10, as depicted in Figure 11 (d). However, this setting significantly compromises benign accuracy, rendering global model unusable. This is because the poisoned data shares similar hidden features as the clean data with the target label, allowing FTA to reuse benign neurons for the backdoor task. Specifically, the baseline attack heavily depends on certain fine-tuned malicious parameters, making it susceptible to Pruning based on the activation of clean data. In contrast, FTA does not rely on the fine-tuned malicious parameters too much.

### A.5    BENIGN ACCURACY OF FTA

We showcase the benign accuracy of both the baseline attack and FTA, and also consider the accuracy without backdoor attacks under `FedAvg`. We start FTA and the baseline from a specific round (e.g., 0 or 200 for different datasets) and perform the attacks during Attack_num rounds. We record the accuracy once the attacks have ended. From Table 2, it is evident that FTA results in a slightly smaller decrease in the benign accuracy compared to baseline attack.

Table 2: Benign accuracy of the baseline attack. FTA and no attackers circumstance under different datasets. Benign accuracy drops by $\leq 1.5\%$ in FTA compared to the accuracy without attack.

| Dataset | Attack start epoch | Attack_num | No attack (%) | Baseline attack (%) | FTA (%) |
|---|---|---|---|---|---|
| Fashion-MNIST | 0 | 50 | 90.21 | 85.14 | 90.02 |
| FEMNIST | 200 | 50 | 92.06 | 91.27 | 92.05 |
| CIFAR-10 | 0 | 100 | 61.73 | 56.34 | 60.61 |
| Tiny-ImageNet | 0 | 100 | 25.21 | 19.06 | 25.13 |

## A.6 THE STRUCTURE OF OUR MODELS

### A.6.1 THE STRUCTURE OF CLASSIFICATION MODELS FOR FASHION-MNIST AND FEMNIST

We use an 8-layer classic CNN architecture for training Fashion-MNIST and FEMNIST datasets. The details are shown in Table 3.

Table 3: The structure of classic CNN model.

| Parameters | Shape | Hyperparameters of layer |
|---|---|---|
| Conv2d | 1*32*3*3 | stride = (1, 1) |
| GroupNorm | 32*32 | eps = $10^{-5}$ |
| Conv2d | 32*64*3*3 | stride = (1, 1) |
| GroupNorm | 32*64 | eps = $10^{-5}$ |
| Dropout2d | | p = 0.25 |
| Linear | 9216*128 | bias = True |
| Linear(For Fashion-MNIST) | 128*10 | bias = True |
| Linear(For FEMNIST) | 128*62 | bias = True |

### A.6.2 THE STRUCTURE OF TRIGGER GENERATOR

In the FTA framework, the trigger generator plays a crucial role in feature extraction in the sense that it aims to align the hidden features of poisoned samples with the target label samples. We utilize the Autoencoder as the trigger generator due to its ability to capture essential features of input and generate outputs satisfying our needs. Moreover, we find that U-Net exhibits comparable performance for trigger generation while requiring less training data, as stated in (Doan et al., 2021b). Therefore, we include U-Net in our experiments. Both U-Net and autoencoder architectures used to train the trigger generator $g_\xi$ are similar to those presented in (Doan et al., 2021b).

### A.6.3 THE STRUCTURE OF CLASSIFICATION MODELS FOR CIFAR-10 AND TINY-IMAGENET

We use a similar ResNet-18 architecture as in (Xie et al., 2019) for training CIFAR-10 and Tiny-ImageNet.

## A.7 OTHER EXPERIMENT SETTINGS

The implementation of all the compared attacks and FL framework are based on PyTorch (Paszke et al., 2019). We test the experiments on a server with one Intel Xeon E5-2620 CPU and one NVIDIA A40 GPU with 32G RAM.

In Fashion-MNIST, CIFAR-10 and Tiny-ImageNet, a Dirichlet distribution is used to divide training data for the number of total agent parties, and the hyperparameter for distribution is 0.7 for the datasets. For FEMNIST, we randomly choose data of 3000 users from the dataset and randomly distribute every training agent with the training data from 3 users. All parties use SGD as an optimizer

and train for local training epochs with a batch size of 256. A global model is shared by all agents, and updates of 10 agents will be selected for aggregating the global model. Benign agents train with a benign learning rate for benign epochs. The attacker's local training dataset is mixed with 80% correct labeled data and 20% poisoned data. The target labels are "sneaker" in Fashion-MNIST, "digit 1" in FEMNIST, "truck" in CIFAR-10 and "tree frog" in Tiny-ImageNet. The attacker has its own local malicious learning rate and epochs to maximize its backdoor performance. It also needs to train its local trigger generator with learning rate and epochs before performing local malicious training on the downloaded global model.

Regarding the attack methods, we set the top-$k$ ratio of 0.95 for Neurotoxin, in line with the recommended settings in (Zhang et al., 2022b). For DBA, we use 4 distributed strips as backdoor trigger patterns. Both the baseline attack and Neurotoxin employ a "square" trigger pattern on the top left as the backdoor trigger. We conduct Edge-case attack on CIFAR-10 and FEMNIST. Specifically, for CIFAR-10, we use the southwest airplane as the backdoored images and set the target label as "bird". For FEMNIST, we use images of "7" in ARDIS (Kusetogullari et al., 2020) as poisoned samples with the target label set as the digit "1". The dataset settings of the experiments are the same as those used in (Wang et al., 2020).

### A.7.1 VISUALIZATION OF DIFFERENT TRIGGER SIZES

The benign and poisoned samples with flexible triggers of different sizes generated by FTA are presented in Figure 12. For Tiny-ImageNet and CIFAR-10, it is hard for human inspection to immediately identify the triggers, which proves the stealthiness in **P3**. In Fashion-MNIST and FEMNIST, the triggers are easier to distinguish because there is only one channel of the input samples in the datasets. But those flexible triggers are still much more stealthy compared to those produced by prior attacks on FL (see Figure 2).

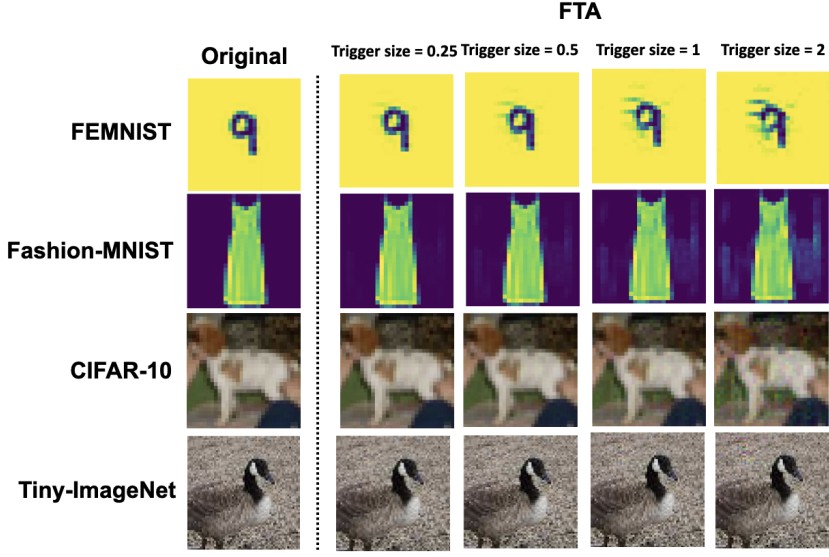

Figure 12: Visualization of backdoored images of different trigger sizes.

### A.8 ABLATION STUDIES IN FTA ATTACK

**Trigger Size.** This size refers to the $l_2$-norm bound of the trigger generated by the generator, corresponding to $\epsilon$ in Algorithm 1. If the size is set too large, the poisoned image can be easily distinguished (i.e., no stealthiness) by human inspection in test/evaluation stage. On the other hand, if we set it too small, the trigger will have a low proportion of features in the input domain. In this sense, the global model will encounter difficulty in catching and learning these features of trigger pattern, resulting in a drop of attack performance.

In Figure 13 (a)-(d), the trigger size significantly influences the attack performance in all the tasks. The accuracies of FTA drop seriously and eventually reach closely to 0% while we keep decreasing the size of the trigger, in which evidences can be seen in CIFAR-10, FEMNIST, and Tiny-ImageNet.

The sample-specific trigger with $l_2$-norm bound of 2 in CIFAR-10 and Tiny-ImageNet is indistinguishable from human inspection (see Figure 12 in Appendix A.7.1), while for Fashion-MNIST and FEMNIST (images with back-and-white backgrounds), additional noise can be still easily detected. Thus, a balance between visual stealthiness and effectiveness should be considered before conducting an FTA.

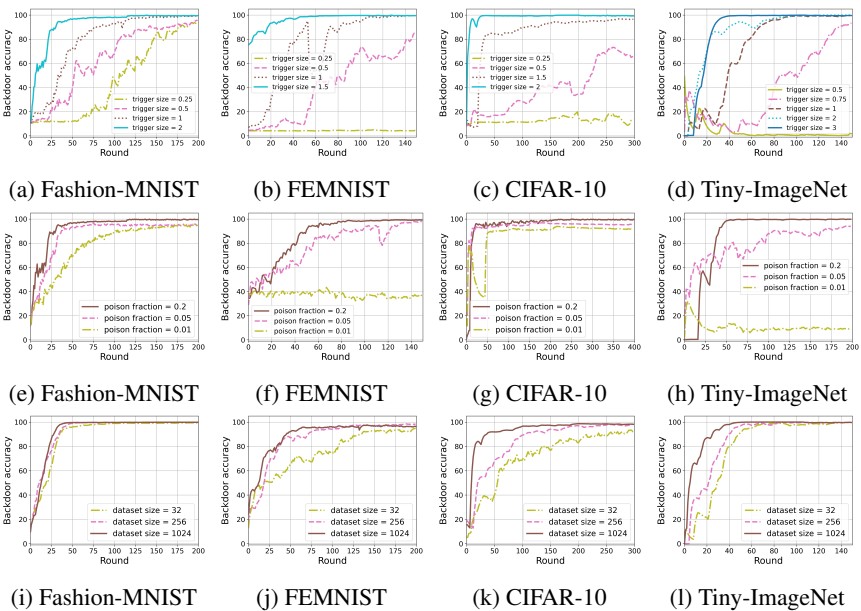

Figure 13: Different hyperparameters on backdoor accuracy. (a)-(d): trigger size; (e)-(h): poison fraction; (i)-(l): dataset size of trigger generator.

**Poison Fraction.** This is the fraction of poisoned training samples in the training dataset of the attacker. Setting a low poison fraction can benefit the attack's stealthiness by having less abnormality in parameters and less influence on benign tasks. But this can slow down the attack effectiveness, as a side effect. Fortunately, we find that FTA can still take effect under a low poison fraction. We set the local training batch size to 256 for all the tasks, follow the standard settings of other FL frameworks, and set the poison fraction as 0.2. As stated in Appendix A.5, this fraction setting cannot degrade the performance of benign accuracy and meanwhile, we would like to explore further to examine the lower bound of the fraction which FTA's performance can tolerate. In Figure 13 (e)-(h), FTA is still effective whilst the fraction drops to 0.05. We also find that sensitivities to poison fraction can vary among tasks. In Fashion-MNIST and CIFAR-10, FTA remains its performance even if poison fraction = 0.01, in which only 3 samples are posioned in each batch. As for FEMNIST and Tiny-ImageNet, under the same rate, the backdoor tasks are dramatically weakened by the benign ones.

**Dataset Size of Trigger Generator.** Theoretically, if this dataset is small-scale, the trigger generator could not be properly trained, thus resulting in bad quality and further endangering the attack performance. During the training, if the attacker controls multiple agents, it can merge all local datasets into one for generator training. However, in many cases, the attacker can only control relatively limited agents and is provided by a small-scale dataset for training. Recall that in Algorithm 1 we use the same dataset for the malicious model and trigger generator training. We set the size of dataset for learning trigger generator to 1024 for all tasks in default. From Figure 13 (i)-(l) in Appendix A.8, we see that this concern should not be crucial for FTA. Even if the size of the dataset is only set to 32, FTA can provide a high attack performance. We note that the training process here is somewhat similar to generative adversarial networks, in which we do not require a large amount of samples in the training dataset.

Table 4: Experimental results on trigger stealthiness (SSIM↑ and LPIPS↓).

| Dataset | Metric | Baseline | DBA | Neurotoxin | Edge-case | FTA(Ours) |
|---|---|---|---|---|---|---|
| Fashion-MNIST | SSIM | 0.9376 | 0.9052 | 0.9359 | - | **0.9967** |
| | LPIPS | NA | NA | NA | NA | NA |
| CIFAR-10 | SSIM | 0.9612 | 0.9440 | 0.9638 | 0.7354 | **0.9978** |
| | LPIPS | 0.0058 | 0.0091 | 0.0075 | 0.3171 | **0.0008** |
| Tiny-ImageNet | SSIM | 0.9851 | 0.9734 | 0.9810 | - | **0.9881** |
| | LPIPS | 0.0072 | 0.0149 | 0.0086 | - | **0.0029** |

Table 5: Time consumption and computational cost (MEAN±SD) of different attack methods in one FL iteration under Fashion-MNIST, CIFAR-10 and Tiny-ImageNet.

| Dataset→ | Fashion-MNIST | | CIFAR-10 | | Tiny-ImageNet | |
|---|---|---|---|---|---|---|
| Attack↓ | Time (s) | Memories (MB) | Time (s) | Memories (MB) | Time (s) | Memories (MB) |
| Benign | 1.62±0.19 | 76.8 | 14.11±1.45 | 125.1 | 37.92±2.71 | 233.5 |
| Baseline Attack | 1.67±0.25 | 81.6 | 14.81±2.10 | 127.2 | 38.52±2.19 | 226.4 |
| DBA | 1.57±0.31 | 81.7 | 14.91±1.86 | 124.7 | 38.74±1.92 | 248.5 |
| Neurotoxin | 3.39±0.66 | 120.4 | 27.85±1.74 | 279.3 | 76.38±3.46 | 478.7 |
| FTA | 2.04±0.52 | 86 | 18.38±1.89 | 169.1 | 46.98±2.14 | 298.4 |

## A.9 NATURAL STEALTHINESS

For each dataset, we randomly select 500 sample images from test dataset to evaluate the trigger stealthiness. As the SSIM value increases, the poisoned sample looks more stealthy. But for LPIPS, that is the other way round. Table 4 shows that FTA achieves excellent stealthiness in all cases. Specifically, SSIM values of FTA are the highest in these datasets, which are close to 1. LPIPS values of FTA are 2-7× improvement to that of baseline attack. Although the baseline attack and Neurotoxin, which uses a universal square pattern, performs well on more complex datasets, using such a patch-based pattern can make the original image look "unnatural".

## A.10 COMPUTATIONAL COST

We understand the significance of the external computational cost and time consumption of backdoor training on malicious devices in our proposed attack under FL scenario. Training with GANs in federated systems introduces extra time consumption. However, our attack does not significantly increase the computational and time cost due to our optimization procedure. Compared to training benign task and baseline backdoor task, FTA only needs to train an additional trigger generator which is actually a small generative neural network. Our generator only consists of several convolutional layers in total. It is worth noting that the datasets used to train both two network structures comprise only 1024 poisoned samples as shown in Table 1 whose size are relatively small compared to the entire training dataset. For instance, the training dataset for our trigger generator accounts for approximately $0.14\%$ of the FEMNIST dataset Therefore, the time consumption and computational cost for training this generative network are very minimal. The remaining time consumption is comparable to training a benign local model. As shown in Table 5, Neurotoxin requires approximately 2× the time and memory compared to benign training to complete backdoor training for one FL round. This is attributed to an additional local benign training requirement in Neurotoxin. However, FTA consumes less than 30% additional time and 25% additional computational cost for backdoor training compared to benign ones. Given that FTA remains under 70% of the cost of Neurotoxin, it is practical to conduct an FTA attack in decentralized scenario.

## A.11 THE IMPORTANCE OF ADAPTABILITY OF FTA TRIGGER GENERATOR

We understand and verify the significance of adaptability in our proposed attack. Figure 14 (a)-(b) demonstrate that the non-adaptive generator provides a lower convergence speed, approximately 70 rounds to achieve 96% backdoor accuracy under CIFAR-10, compared to the adaptive variant. Also, as shown in Figure 14 (c)-(d), the non-adaptive variant cannot evade norm clipping (vector-wise scaling) since it needs to significantly tune malicious parameters to achieve high backdoor accuracy. Our findings provide evidence that the adaptive variant indeed enhances the stealthiness of FTA in the parameter space.

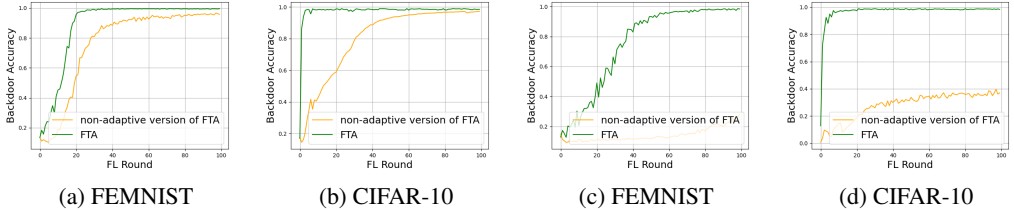

(a) FEMNIST     (b) CIFAR-10     (c) FEMNIST     (d) CIFAR-10

Figure 14: (a)-(b): The comparison between FTA and its restricted version under no defense in FEMNIST and CIFAR-10. (c)-(d): The comparison between FTA and its restricted version under norm clipping in FEMNIST and CIFAR-10.

## A.12 DISCUSSION

In this work, we concentrate on the computer vision tasks, which have been the focus of numerous existing works (Xie et al., 2019; Wang et al., 2020; Doan et al., 2021b; Zhao et al., 2022b; Ozdayi et al., 2021). In the future, we intend to expand the scope of this work by applying our design to other real-world applications, such as natural language processing (NLP) and reinforcement learning (RL), as well as other vision tasks, e.g., object detection.

The primary focus of FTA is to achieve stealthiness rather than durability, in contrast to other attacks such as Neurotoxin (Zhang et al., 2022b). Neurotoxin manipulates malicious parameters based on gradients in magnitude, which yields a clear increase in the dissimilarity of parameters and thus harms the stealthiness of the attack. FTA addresses the dissimilarity difference of weights/biases introduced by backdoor training by using a stealthy and adaptive trigger generator, which makes the hidden features of poisoned samples similar to benign ones. We emphasize that the durability of backdoor attacks on FL is orthogonal to the main focus of this work, and we leave it as an open problem. A possible solution to achieve persistence could be to decelerate the learning rate of malicious agents, as proposed in (Bagdasaryan et al., 2020).

**Comparison.** In addition to the defenses evaluated in this paper, we discuss our attack effectiveness under other defenses below. As depicted in FLDetector (Zhang et al., 2022a), in a typical FL scenario where the server does not have a validation dataset that Fltrust (Cao et al., 2021) requires, the global model remains susceptible to backdoor attacks. However, the stringent demand by FLtrust for an extra validation dataset could not be practical for conventional FL frameworks and applications. Furthermore, Fltrust eliminates backdoor effectiveness based on cosine dissimilarity which is similar to the approach used in FLAME. As shown in Figure 5, FTA's malicious updates have less dissimilarity to benign updates than the baseline attack's. Therefore, we can state that FTA can evade Fltrust according to the results obtained under FLAME. DnC (Shejwalkar & Houmansadr, 2021) primarily focuses on untargeted poisoning attacks rather than backdoor attacks, and its main objective is to reduce the accuracy of FL models. Accordingly, we do not consider it as a "proper" SOTA backdoor defense (to our attack). In particular, DnC is a kind of vector-wise filtering defense. In our experiments, conducted under Multi-krum and RFA, we ascertain that FTA is robust against vector-wise filtering. In conclusion, FTA can also successfully evade DnC, much like Multi-krum and RFA. As for certified defense like Flcert (Cao et al., 2022), while it is a promising approach to robustness certification, it is not intended to detect and filter out malicious updates in FL. As outlined in Flcert, the certified accuracy of the global model experiences a decline with the increase of malicious agents. Fortunately, FTA can cope with a very challenging threat model, where the attacker is allowed to control merely one malicious agent out of thousands. We thus can achieve a certified accuracy almost on par with the original global model accuracy against Flcert. FLIP (Zhang et al., 2023) only considers static backdoors as potential attacks, i.e. patch-based patterns, whereas FTA can use flexible triggers to break FLIP's threat model. Using the flexible trigger generator, FTA can produce sample-specific triggers which pose challenges when applying universal trigger inversion method in FLIP's step 1.

