# OpenReview forum: "FTA: Stealthy and Adaptive Backdoor Attack with Flexible Triggers on Federated Learning"
_ICLR.cc/2024/Conference — Submitted to ICLR 2024_

### Official Review · Reviewer_cGsZ · 2023-10-26

**Soundness:** 3 good
**Presentation:** 2 fair
**Contribution:** 2 fair
**Rating:** 5
**Confidence:** 4

**Summary:**

This paper proposes a stealthy and adaptive backdoor attack with flexible triggers for federated learning.

**Strengths:**

The studied problem of backdoor attack in federated learning is important.

The experiment results show that the generated trigger is less perceptible in human eyes, and comprehensive experiments are done to verify the success of the attack.

**Weaknesses:**

1. The novelty of the formulated problem, as well as the method design is limited.  Specifically, the problem formulation in Eq. (1)  is mostly the same with Eq. (3) of Lira [A], except for some minor differences like separating poisoned and clean datasets.  The solution to solve the proposed problem is also quite standard by alternating optimization of the two variables, which is also adopted by  Wasserstein Backdoor [B]. The generator of the trigger is also following the autoencoder structure as adopted by [A].   In this sense, the proposed attack seems to be a direct migration of Lira into a federated learning setting, which looks quite incremental.

2. The defense baselines are not comprehensive. The authors can consider adding more defense baselines, e.g., RLR [C], Crfl [D],  to show that the attack can successfully break through defenses other than cluster-based filtering.

3. It is unclear why optimizing the triggers can guarantee a better attack towards cluster-based filtering (or minimizing the distance of updates with the poisoned update), as this is not reflected in the problem formulation. See details in my questions part.

4. There are some issues with the experiment results and the setup. The baseline benign accuracy is very low (shown in Table 2, 61.73% benign accuracy for CIFAR10 with ResNet, and also low for TinyImagNet), which makes the correctness of the experiment implementation questionable.  The setup of local epochs is also strange, in that the malicious clients run more epochs than the benign clients. This might introduce bias to other baselines because this would make the malicious updates significantly larger than other benign updates, which may affect the performance of other attack baselines when against filtering-based defense.   Also, the authors should test the results in IID setting as well as various Non-IID parameters to show its effectiveness.


[A] Doan K, Lao Y, Zhao W, et al. Lira: Learnable, imperceptible and robust backdoor attacks[C]//Proceedings of the IEEE/CVF international conference on computer vision. 2021: 11966-11976.

[B] Doan K, Lao Y, Li P. Backdoor attack with imperceptible input and latent modification[J]. Advances in Neural Information Processing Systems, 2021, 34: 18944-18957.

[C] Ozdayi M S, Kantarcioglu M, Gel Y R. Defending against backdoors in federated learning with robust learning rate[C]//Proceedings of the AAAI Conference on Artificial Intelligence. 2021, 35(10): 9268-9276.

[D] Xie C, Chen M, Chen P Y, et al. Crfl: Certifiably robust federated learning against backdoor attacks[C]//International Conference on Machine Learning. PMLR, 2021: 11372-11382.

**Questions:**

It is suggested in Section 3.3 "one may consider alternately updating fθ while keeping Tξ unchanged, or the other way round... (but this couldn't work well)". However, it is later claimed that "Inspired by (Doan et al., 2022), we divide local malicious training into two phases. In
phase one, we fix the classification model fθ and only learn the trigger function Tξ. In phase two, we use the pre-trained Tξ∗ to generate the poisoned dataset and train the malicious classifier fθ". In my understanding, the two descriptions of alternating optimization are identical. Can the authors elaborate on it?

It is claimed on page 4 that "A stealthy backdoor attack on FL should mitigate the routing introduced by backdoor task and guarantee the stealthiness of model parameters instead of just the hidden features of poisoned samples compared to their original inputs". However, it is unknown how the authors are achieving this goal with their problem formulation in Eq. (1).

---

> ### Author Response · Authors · 2023-11-16
> **Thank you for your elaborate review**
>
> Thank you for your detailed and valuable comments.
>
> ***Weakness 1***: Solving bi-level problems by alternating optimization of the two variables is commonly used by Lira, Wasserstein Backdoor (WB), DEFEAT [A] and Marksman [B].
>
> But our novelty lies in two main aspects:
>
> 1.  ***Motivation***:
> Most generator based attacks, such as Lira and WB, focus on stealthiness in the input/feature space. However, we consider attack stealthiness in the parameter space, posing security challenges for robust FL frameworks.
>
> 2.  ***Technical contribution***:
> We poison *iterative* global models across FL rounds rather than *universal* classification model such as eq.(3) in Lira. We achieve considerable practical time consumption of training trigger generators in FL by learning in an incremental manner with fewer epochs.
> We also introduce a novel discovery derived from eq.(1) of FTA that by learning poisoned data (containing flexible triggers) with target label, we can make similar features shared by clean data of target label and thus achieve stealthiness in the parameter space.
>
> To clarify the differences between existing methods and FTA, we have refined Eq.(1) and Alg.1 in Section 3.1.
>
> ***Weakness 2***:
> We test FTA under 9 SOTA robust FL defenses based on 5 mechanisms including vector-wise scaling, cluster-based filtering, vector-wise filtering, dimension-wise filtering, post-training and sparsification, as shown in Fig.4 in Sec.4.3 and Fig.7-10 in Appendix A.4.
> As for certified defense, while it is a promising approach to robustness certification, as described in CRFL, it is not designed to detect and filter out malicious updates in FL. As a result, we did not consider using certified FL frameworks (as strong defenses) for comparison. However, RLR can effectively filter out malicious updates, we have tested FTA under RLR as the variant of SignSGD.
> As shown in Fig.11 (a)-(b) in Appendix A.4.5, FTA achieves above 98\% backdoor accuracy in both i.i.d and non-i.i.d manners.
>
> ***Weakness 3***:
> By learning poisoned data with flexible triggers to the target class, we achieve stealthiness in parameter space with the formula (i) of eq.(1) since it enables the similarity of hidden features between poisoned and clean data with target label as shown in Fig.5 in Sec.4.4.
> This (similarity) allows FTA to reuse benign neurons for backdoor tasks. So FTA does not need to significantly fine-tune malicious parameters. Due to the imperceptible pertubation of parameters, FTA can evade the cluster-based defenses based on distance metrics such as cosine distance and $L_2$-norm distance.
>
> ***Weakness 4***:
> 1.  Accuracy:
> Global model fail to achieve a satisfactory accuracy because: (a) The performance of FL is naturally worse than centralized machine learning due to local data distribution; (b) We attack the global model which is learned from scratch. The reason is also mentioned in DBA [D] - benign updates share common patterns of gradients and have a larger magnitude than malicious updates, which restricts attack effectiveness. We consider attacks from scratch as a more challenging scenario than attacks after model convergence.
>
> 2.  Epochs:
> Sorry for the confusion.
> Table 1 presents the default hyperparameter setting without defense.
> Assuming that defense is not considered, applying more backdoor epochs enhances the effectiveness of other attacks, mentioned in Weakness 4.1.
> When considering defense mechanisms, we use the same number of malicious epochs as benign ones for a fair comparison.
>
> 3.  Non-i.i.d:
> Following RLR, we test non-i.i.d setting in FEMNIST which is a benchmark dataset under SOTA defenses as shown in Fig.3(b) in Sec.4.2. Our results confirm that attack effectiveness is not harmed in practical non-i.i.d. distributions. This is so because FTA can always learn the generator from malicious local data. We can conduct extra experiments but the conclusion remains the same.
>
> ***Question 1***: Sorry for the confusion.
> Most function based attacks alternately update $f_\theta$ while keeping $\mathcal{T}_\xi$ unchanged, or the other way round, for many iterations (usually $>1$). Inspired by (Doan et al., 2022), one local FL round of FTA is divided into two phases, and each phase is executed for only one iteration with fewer epochs. Consequently, the two descriptions are different in terms of learning process.
>
> We've further clarified the differences between existing methods and ours in the revised manuscript, please see the first paragraph of Section 3.3 for more details.
>
> ***Question 2***: Please refer to W3.
>
> [A]Zhendong Zhao, et al. DEFEAT: Deep Hidden Feature Backdoor Attacks by Imperceptible Perturbation and Latent Representation Constraints. CVPR'22. [B]Doan, Khoa D., et al. Marksman backdoor: Backdoor attacks with arbitrary target class. NeurIPS'22. [C]Wu, Chen, et al. Mitigating backdoor attacks in federated learning. arXiv'20. [D]Xie, Chulin, et al. Dba: Distributed backdoor attacks against federated learning. ICLR'19.

---

> > ### Comment · Reviewer_cGsZ · 2023-11-20
> > **Thanks for the clarification, but my concern remains.**
> >
> > 1. In the rebuttal, the authors list two main novelties compared to Lira, Wasserstein Backdoor (WB), DEFEAT [A] and Marksman [B]. However, they did not convince me.
> >
> > * "Attack stealthiness in the parameter space". This is unclear to me, as how to achieve attack stealthiness does not seem to be reflected in the loss function Eq.(1). To me, the defined loss function is the same with Eq (3) in Lira, except that the authors separate clean and poisoned dataset.
> >
> >  * "We poison iterative global models across FL rounds rather than universal classification model". I couldn't understand what it meant by "poisoning universal classification model". Lira is also poisoning model iterate, but of course not a global model because they are not doing it in a federated setting. The only difference in optimization is that Lira is also updating the trigger generator in the first stage, but the authors fix it in FTA.
> >
> >
> > 2. Please don't involve $t$ in the loss function definition in Eq.(1) in order to distinguish your problem from Lira. The revised problem is not well-defined after adding $t$ to it.
> >
> >
> > 3. What is the meaning of RLR as the variant of SignSGD? As far as I know, vanilla RLR does not use SignSGD.
> >
> > The main weakness of this paper is the novelty, which is also pointed out by Reviewer gADe.  Though the attack method looks technically sound, I would say that the novelty and technical contribution is not enough, at least not enough for a top venue like ICLR. I would keep my borderline rejection score, but would probably change my score if AC or other reviewers support you.

---

> > > ### Author Response · Authors · 2023-11-20
> > > **Thanks for your valuable comments.**
> > >
> > > Thanks for sharing your comments.
> > > While appreciating your perspective, we respectfully disagree with your concern regarding the novelty.
> > >
> > > We acknowledge that our equation bears similarities to Lira.
> > > BUT it's important to note that our novelty does not stem from the learning objective.
> > >
> > > > 1.1 This is unclear to me, as how to achieve attack stealthiness does not seem to be reflected in the loss function Eq.(1).
> > >
> > > We present the new discovery as shown in Figure 5 (a)-(b).
> > > As addressed in the rebuttal (Weakness 3), by learning the formula (i), FTA enables the similarity of hidden features between poisoned and clean data.
> > > This can make the poisoned model NOT to be tuned maliciously in parameter space - as the poisoned data "looks like" clean one.
> > >
> > > > 1.2 I couldn't understand what it meant by "poisoning universal classification model...The only difference in optimization is that Lira is also updating the trigger generator in the first stage, but the authors fix it in FTA.
> > >
> > > Centralized attacks (such as Lira) concentrate on a static scenario where the target model remains unaltered by by others during poisoning.
> > > IN CONTRAST, our target model undergoes dynamic changes orchestrated by the *server*, driven by benign updates in each FL round.
> > > This renders the equation derived from a centralized scenario inapplicable to FL.
> > > Furthermore, in the static scenario, Lira can learn an optimal trigger generator and conclude malicious training when the attack achieves a satisfactory backdoor accuracy.
> > > BUT, in a decentralized setting, it is impractical to train an optimal generator in each FL round, primarily due to time constraints.
> > >
> > > > 2 Please don't involve $t$ in the loss function definition in Eq.(1) in order to distinguish your problem from Lira. The revised problem is not well-defined after adding $t$ to it.
> > >
> > > $t$ remarks a crucial factor in our scenario, given that our target classification model in Eq.(1) differs significantly from that of Lira.
> > > In fact, the changing of global model in our scenario poses a more formidable challenge compared to the centralized setting.
> > > For further clarification, please refer to the preceding response outlining the distinctions between our approach and Lira.
> > >
> > > > 3 What is the meaning of RLR as the variant of SignSGD? As far as I know, vanilla RLR does not use SignSGD.
> > >
> > > RLR employs a similar strategy to SignSGD for aggregating updates, specifically by considering the signs of the gradients.
> > > The key distinction lies in:  RLR compares the absolute value of the sum of signs of updates to a threshold in order to determine the final element-wise sign; BUT SignSGD relies on the majority of the sum of signs.
> > > As such, we regard RLR as a variant of SignSGD.
> > >
> > > To our knowledge, this is the first work to apply trigger generator based attack on federated learning.
> > > This novel approach poses a formidable threat to SOTA robust frameworks, including FLIP [A].
> > >
> > > We trust that our clarification has effectively conveyed the novelty of our work.
> > > If there are any further questions or aspects requiring additional explanation, please do let us know.
> > >
> > > [A]Kaiyuan Zhang, et al. Flip: A provable defense framework for backdoor mitigation in federated learning. ICLR'23.

---

> ### Comment · Reviewer_cGsZ · 2023-11-20
> **Why you don't directly test RLR but test a modified version of RLR?**
>
> Thanks for the clarification. I am just wondering why you don't directly test RLR but SignSGD? I do think that there is a dinstinction between them.

---

> > ### Author Response · Authors · 2023-11-20
> > **Thanks for your prompt reply, we have tested RLR to verify our attack effectiveness.**
> >
> > Thanks for your reply. We have evaluated our attack effectiveness under RLR on FEMNIST and Tiny-ImageNet. As shown in Figure 11 (a)-(b) in Appendix A.4.5 (page 18), FTA achieves above 98% backdoor accuracy in both i.i.d and non-i.i.d manners. Please also refer to Weakness 2 in the rebuttal.

---

### Official Review · Reviewer_gADe · 2023-10-31

**Soundness:** 3 good
**Presentation:** 3 good
**Contribution:** 2 fair
**Rating:** 5
**Confidence:** 4

**Summary:**

This paper proposes a backdoor attack in the federated learning scenario, using a generative model that optimizes the perturbation to achieve stealthiness. The trigger is optimized during the training to be flexible and adaptive. The evaluation shows that it can achieve a 98% attack success rate.

**Strengths:**

The paper focuses on an important problem, and the solution is clear. It is easy
to follow and understand.

The evaluation uses multiple datasets and models, also compares with multiple
baselines.

**Weaknesses:**

The core idea of the paper is to leverage a generative model to add adaptive
perturbations, which has been studied in many existing works, e.g., Cheng et al.
AAAI 2021, Dynamic attack, etc. The paper applies this idea in the federated
learning domain, but there is nothing that the method is specific to this
domain. Namely, I do not see any challenges because of federated learning that
prevents existing work from being used. Thus, I do not think the paper is novel.

Related to the previous question, there has been studies in detecting function
based attacks, and the paper does not discuss that.

**Questions:**

What is the main technical contribution of the paper?

---

> ### Author Response · Authors · 2023-11-16
> **Response to Reviewer gADe**
>
> Many thanks for reviewing our paper and highlighting your confusion.
>
> It is undeniable that generative models have been applied to produce triggers for backdoor attacks in centralized machine learning.
>
> But there are two main ***differences*** between these attacks and ours:
>
> 1.  Motivation:
> Most trigger generator based attacks focus on the stealthiness in the *input* and *latent feature* space. For example, Cheng et al. (AAAI 2021) aims to implant style-based triggers that look natural and Dynamic attack plants backdoors with patch-based triggers randomly in the input space. However, our emphasis is on achieving the stealthiness in the *parameter* space while maintaining the stealthiness in the input space. Specifically, our work concentrates on ensuring the similarity of latent representations between poisoned and clean data with target label.
>
> 2.  Technical difficulties:
>
>     - Decentralized v.s. Centralized Scenarios: Different from function based attacks that aim to poison a *universal* classification model, our approach considers *iterative* global models across FL rounds.
>
>     - Time consumption: The additional time consumption caused by updating trigger generator from scratch in centralized setting is impractical for FL due to the restricted aggregation time on server.
>
> We, for the first time, design a trigger generator for backdoor attacks against a decentralized setup, bridging the above gap with the following technical ***contributions***:
>
> 1.  To guarantee stealthiness in the parameter space under FL and prevent malicious clients from being detected, our trigger generator ensures the similarity of hidden features between clean data of target class and poisoned data.
>
> 2.  We customize our learning process of generator in an incremental manner to adapt to the updating of global model, allowing fewer epochs in local backdoor training. Thanks to our advanced trigger-generator training process, malicious clients can send their malicious updates to server within an acceptable time frame, hereby ensuring attack effectiveness.
>
> Detection considering function based attacks, such as [A], cannot identify outliers in the parameter space of FTA.
> As shown in Fig.5(b), FTA achieves a similar hidden feature of poisoned data as that of clean data belong to the target class. Therefore, it can use clean neurons for backdoor tasks without inducing anomaly to backdoor neurons.
> We can further perform additional experiments to demonstrate attack effectiveness under function based defenses, but this will not impact our overall conclusion.
>
> To further clarify the above points, we've made the following improvements in our revised manuscript:
> (1) We've reviewed the literature on the use of trigger generators in centralized ML, in related work;
> (2) We've elaborated on the challenges/difficulties that we overcome in applying existing trigger generators in FL, in our contributions, Sec.1.
>
> [A] Wang, Hang, et al. Universal post-training backdoor detection. arXiv'22.

---

> ### Author Response · Authors · 2023-11-23
> **Thanks for your valuable comments, please let us know if there are any further questions.**
>
> Thanks for your valuable comments. If there are any further questions or aspects requiring additional explanation, please do let us know.

---

### Official Review · Reviewer_nnMX · 2023-10-31

**Soundness:** 3 good
**Presentation:** 2 fair
**Contribution:** 2 fair
**Rating:** 6
**Confidence:** 3

**Summary:**

The authors propose a generator-assisted backdoor attack (FTA) against robust FL. The newly designed generator is flexible and adaptive, where a bi-level optimization problem is formed to find the optimal generator.

**Strengths:**

1. Clear model and algorithm

1. T-SNE visualization of hidden features and similarity comparison are helpful.

**Weaknesses:**

1. To emphasis the importance of flexibility and adaptability, the authors may consider add some experiments compared with their restricted version attacks against fixed batch of data and under non-adaptive setting.

2. The current baselines are all fixed and non-adaptive. I suggest the authors compare their results with SOTA trigger generated based attacks as [1] and [2].

3. The post-training stage defenses play a vital role in countering backdoor attacks. Even within the context of FL, certain techniques such as Neuron Clipping [3] and Pruning [4] have demonstrated their effectiveness in detecting and mitigating the impact of backdoor attacks. Consequently, I am curious to know how the proposed FTA performs when subjected to these post-training stage defenses.


[1] Salem, Ahmed, et al. "Dynamic backdoor attacks against machine learning models." 2022 IEEE 7th European Symposium on Security and Privacy (EuroS&P). IEEE, 2022. [2] Doan, Khoa D., Yingjie Lao, and Ping Li. "Marksman backdoor: Backdoor attacks with arbitrary target class." Advances in Neural Information Processing Systems 35 (2022): 38260-38273. [3] Wang, Hang, et al. "Universal post-training backdoor detection." arXiv preprint arXiv:2205.06900 (2022). [4] Wu, Chen, et al. "Mitigating backdoor attacks in federated learning." arXiv preprint arXiv:2011.01767 (2020).

**Questions:**

1. How to choose/tune a good or even an optimal (is it exist?) $\epsilon$?

2. The structure of generator network is crucial to balance the tradeoff between effectiveness and efficiency since the authors want to achieve flexible (each training example) and adaptive (every FL epoch). In the centralized setting in [1] [2], trigger generators specific to every label need be trained one time before machine learning, and it still require some training time. I wonder is there any modifications the authors made to increase the efficiency of the training to achieve a flexible and adaptive attack?


[1] Doan, Khoa, et al. "Lira: Learnable, imperceptible and robust backdoor attacks." Proceedings of the IEEE/CVF international conference on computer vision. 2021. [2] Doan, Khoa D., Yingjie Lao, and Ping Li. "Marksman backdoor: Backdoor attacks with arbitrary target class." Advances in Neural Information Processing Systems 35 (2022): 38260-38273.

---

> ### Author Response · Authors · 2023-11-16
> **Response to Reviewer nnMX**
>
> We thank the reviewer for the valuable comments. We put the following clarifications to address the concerns.
>
> ***Weakness 1***: We understand and verify the significance of adaptability and flexibility in our proposed attack.
>
> 1.  ***Adaptability.***
> It is essential to perform an adaptive attack due to the continuous update of global model across FL rounds. To validate this, we have introduced an extra comparison by evluating FTA with limited adaptability. Specifically, we have conducted a toy experiment under a non-adaptive setting, training the generator only in the first FL round with/without defenses. The results demonstrate that the adaptive variant indeed enhances the stealthiness of FTA in the parameter space, please see Fig.14 (a)-(b) and Appendix.A.11 for more analysis.
>
>
> 2.  ***Flexibility.*** It refers to the sample-specific characteristic of our trigger generator. This attribute guarantees the similarity of hidden representation between benign and poisoned data, thereby preventing anomaly in the parameter space. To validate the importance of flexibility, we use t-SNE to project the hidden representation of benign data along with poisoned data with/without flexibility, see Fig.5 (a)-(b) in Sec.4.4. This demonstrates that the hidden feature dissimilarity between benign and poisoned data (i.e., poisoned sample with universal patch-based trigger) with "non-flexibility" is larger than that of our flexible version of FTA.
> Therefore, the significance of flexibility is confirmed.
>
> ***Weakness 2:***
> [1][2] are SOTA trigger generator based attacks in a centralized setup. However, several crucial differences make most generator based attacks incomparable to FTA in the FL scenario:
>
> 1.  ***Motivations:*** Current generator based attacks emphasize the stealthiness in the *input/feature* space. In contrast to their motivations, our approach considers attack stealthiness in the *parameter* space, presenting unique security challenges against robust FL frameworks.
>
> 2.  ***Not applicable to FL:*** Directly training existing generator based attacks on FL could incur impractical time consumption.
> Moreover, these attacks typically emphasize the *universal* classification model, whereas FTA specifically focuses on the *iterative* global model.
>
> ***Weakness 3:***
> Post-training stage defenses play crucial roles under robust FL. Hereby, we have evaluated the effectiveness of the baseline attack and FTA under Pruning. We showcase that FTA still maintains outstanding backdoor accuracy against Pruning, please see Appendix.A.4.8 and Fig.11 (c)-(d) for more analysis.
>
> Neuron Clipping detects backdoor based on the observation that backdoor neurons exhibit significantly heightened activations. However, as shown in Fig.5(b), FTA achieves a similar hidden feature (activation) distribution to benign data of the target class. This (similarity) allows FTA to use clean neuron for backdoor tasks without inducing anomaly to backdoor neurons. Therefore, FTA could evade Neuron Clipping. While we can conduct additional experiments as backups, this will not affect our conclusion.
>
> ***Question 1:***
>
> $\epsilon$ balances trigger invisibility and backdoor accuracy, and the optimal $\epsilon$ is subjective to attacker preferences.
> Our goal is to select an $\epsilon$ value that is as small as possible without significantly compromising backdoor accuracy.
> For example, as shown in Fig.13 (a), a value of $\epsilon$ less than 0.5 tends to negatively impact both convergence and attack performance.
> In contrast, values of $\epsilon$ greater than 1 do not significantly contribute to improved backdoor accuracy.
> Therefore, we choose the smallest $\epsilon$, such as $\epsilon=1$ for Fashion-MNIST, ensuring both satisfactory backdoor accuracy and (trigger) invisibility simultaneously.
>
>
> ***Question 2:***
>
> We set a smaller batch size and fewer epochs of our generator to enhance the efficiency of FTA in FL compared to most generated based attacks in a centralized setup.
> This strategy performs well as we tailor the learning process of generator by updating it in an incremental manner across FL rounds to adapt to iterative global model.
> In other words, we update the generator and model only once, avoiding training our generator from scratch in each FL round.
> As shown in Tab.1 in Appendix.A.3, we use only 30 epochs and 256 samples in one FL round, ensuring an effective attack.

---

> > ### Comment · Reviewer_nnMX · 2023-11-23
> >
> > I appreciate the authors' response and informative clarification. After reading the rebuttal and other reviewers' comments, part of my concerns has been addressed.

---

> > > ### Author Response · Authors · 2023-11-23
> > > **Thanks for the detailed reading and valuable comments.**
> > >
> > > Thanks for the detailed reading and valuable comments. If there are any further questions or aspects requiring additional explanation, please do let us know. Thanks again for the appreciation of our work.

---

### Author Response · Authors · 2023-11-16
**Elaboration of Novelty**

Dear reviewers,

Thank you for your time and efforts to facilitate the discussion of our paper.

Our *novelty* relies on the following aspects:

(1) ***Decentralized v.s. Centralized Scenarios.***
Different from current generator based attacks that aim to poison *universal* classification model, FTA considers *iterative* global models across FL rounds. We tailor learning process of generator in a gradual progression to adapt to the update of global model, avoiding the necessity to start the learning progress from scratch in every FL round. This enables fewer epochs in local backdoor training.


(2) ***Motivation.*** The majority of generator based attacks aim to produce imperceptible sample-specific triggers in the *input/feature* space. In contrast, FTA is to not only generate imperceptible triggers but also address the anomaly of malicious updates in FL setups, specifically in the *parameter* space.

(3) ***Methodology.*** Instead of directly imposing constraints on the dissimilarity of malicious and benign updates by cosine or $L_2$-norm distance or restraining the hidden feature representations, we reveal a new finding of our formulation that can naturally eliminate the parameter abnormality mentioned in *P1-2*.

Last but not least, the generator of FTA can produce flexible triggers according to its input.
As a result, FTA can naturally evade the SOTA robust FL defense - FLIP[A], which relies on the universal trigger inversion technique.

In conclusion, FTA poses security challenges for SOTA robust FL frameworks.

Kind regards,

Authors

[A]Kaiyuan Zhang, et al. Flip: A provable defense framework for backdoor mitigation in federated learning. ICLR'23.

---

### Meta-Review · Area_Chair_tgWq · 2023-12-10

**Metareview:**

This work studied the backdoor attack against federated learning. In each round, the trigger generator and the local model are alternatively updated.

The main critical point is the technical novelty, especially its high similarity with lira, including the objective and optimization. The authors claimed three aspects (scenarios, motivation, method). However, they are not very convincing. For example, the authors emphasized the stealthiness in parameter space. However, from the objective and optimization, it is unclear which term leads to this characteristic.

Besides, the reviewers suggested more defense methods, such as post-training defense. The authors added a few, but without clear explanation. Even the stealthiness in parameter space is true, several post-training defenses may still work. There is no evidence to  show that the proposed attack method could evade existing post-training backdoor defenses.


Hope all reviews are helpful to further improve this work.

**Justification For Why Not Higher Score:**

see above

**Justification For Why Not Lower Score:**

n/a

---

### Decision · Program_Chairs · 2024-01-16

Reject